# Orbitofrontal PV interneurons modulate social interaction via default mode network dynamics
Elmira Khatamsaz[1], Tudor M. Ionescu[1], Katja Keppler[1], Franziska Stoller[1], Dennis Kätzel [2], Hugh M. Marston [1] & Bastian Hengerer [1] ✉

Social interactions are critical for mental health and are frequently disrupted in neuropsychiatric disorders. Clinical data suggest a link between dysfunction of the default mode network (DMN) and social impairment. To advance our understanding of the neurobiological mechanisms underlying social dysfunction, we back-translate these clinical findings into a preclinical setup, demonstrating that impaired DMN connectivity - achieved through activation of Parvalbumin interneurons in the orbitofrontal cortex (OFC) reduces normal social behaviour.

Social dysfunction, often manifesting as social withdrawal, is a hallmark of numerous neuropsychiatric and neurological disorders[1]. Despite differing neurobiological underpinnings, both Schizophrenia and Alzheimer's disease (AD) patients exhibit impaired social and behavioral functions, underscoring the importance of transdiagnostic approaches[2]. In the European Union-funded PRISM project (Psychiatric Ratings using Intermediate Stratified Markers), dysconnectivity in the default mode network (DMN) has been identified as a potential endophenotype for social dysfunction[3–7]. However, the potential underlying mechanisms driving DMN dysconnectivity and subsequent social impairment remain unclear.

This study focuses on the orbitofrontal cortex (OFC), a region critical for decision-making and behavior regulation[8,9]. Recent clinical data suggest that greater social support correlates with lower mean diffusivity in white matter tracts, specifically the forceps minor (fmi) and inferior fronto-occipital fasciculus (IFOF), both converging in the OFC. These tracts are critical brain circuits linking key DMN regions, including the medial prefrontal cortex, posterior cingulate cortex, and the OFC, all of which are implicated in social cognition, emotional regulation, and self-referential processing[10]. In mice, the OFC shows strong functional connectivity with regions such as the retrosplenial cortex, ventral hippocampus, thalamus, and prefrontal cortex, and in rats, the subdivisions of the OFC have similarly been shown to integrate within a DMN-like circuit involving the prelimbic cortex, cingulate cortex, posterior parietal cortex, and hippocampus[11,12]. Using a chemogenetic approach, we investigated how modulation of OFC affects DMN connectivity and social behavior. Parvalbumin (PV) interneurons are increasingly recognized as key regulators of social behavior, with recent evidence showing that their activity in regions such as the anterior cingulate cortex significantly shapes social interaction patterns in rodents[13]. We hypothesized that PV interneuron activation in the OFC of healthy mice would impair DMN connectivity and that decreased DMN connectivity correlates with impaired social behavior.

## Results and discussion

To test this hypothesis, PV-Cre mice and the excitatory designer receptor exclusively activated by designer drugs (DREADD) hM3Dq were used to selectively modulate PV interneurons in the OFC (Figs. 1a and S1a)[14].

Chemogenetic activation of PV interneurons in the OFC has been verified by immunohistochemistry against the immediate early gene c-fos, a surrogate marker for neuronal activity (Fig. S1b–d and Methods)[15,16]. To assess the effects of activation of PV interneurons in the OFC, we recorded global brain activity and functional connectivity (FC) with high-resolution functional ultrasound (fUS) 3 weeks post-surgery (Fig. 1a and Methods). Cerebral blood volume (CBV) values in 29 predetermined regions of interest (ROIs) were measured before and after CNO administration, with relative changes expressed as percentage change from baseline. The results, shown in three multi-slice images with specific thresholds (Fig. 1c, d), revealed robust increases in CBV within four regions selectively in the hM3Dq group: prelimbic cortex (PreL), OFC, primary and secondary motor cortex (M1, M2) (Fig. 1d). Hemispheric analysis of these regions further confirmed the increases (Fig. S8a–d). Spearman's rank correlation demonstrated dynamic CBV coupling between the OFC and Insula (Ins) in the hM3Dq group (Fig. 1e, f).

For FC analysis, whole-brain correlations across 29 ROIs were computed (Fig. S9). Focusing on the DMN, z-transformed correlation coefficients of fUS signal time courses revealed that acute CNO administration in the hM3Dq group significantly reduced FC between key DMN regions, particularly with the dorsal hippocampus (CAd), subiculum (SUB), thalamus (Th), retrosplenial (RSP), and cingulate (ACg) cortex (Fig. 2a, b). Three CAd connections (with SUB, somatosensory area (SS), and Th), two further

[1]Boehringer Ingelheim Pharma GmbH & Co. KG, Biberach, Germany. [2]Institute for Applied Physiology, Ulm University, Ulm, Germany.
✉e-mail: bastian.hengerer@port.lukasiewicz.gov.pl

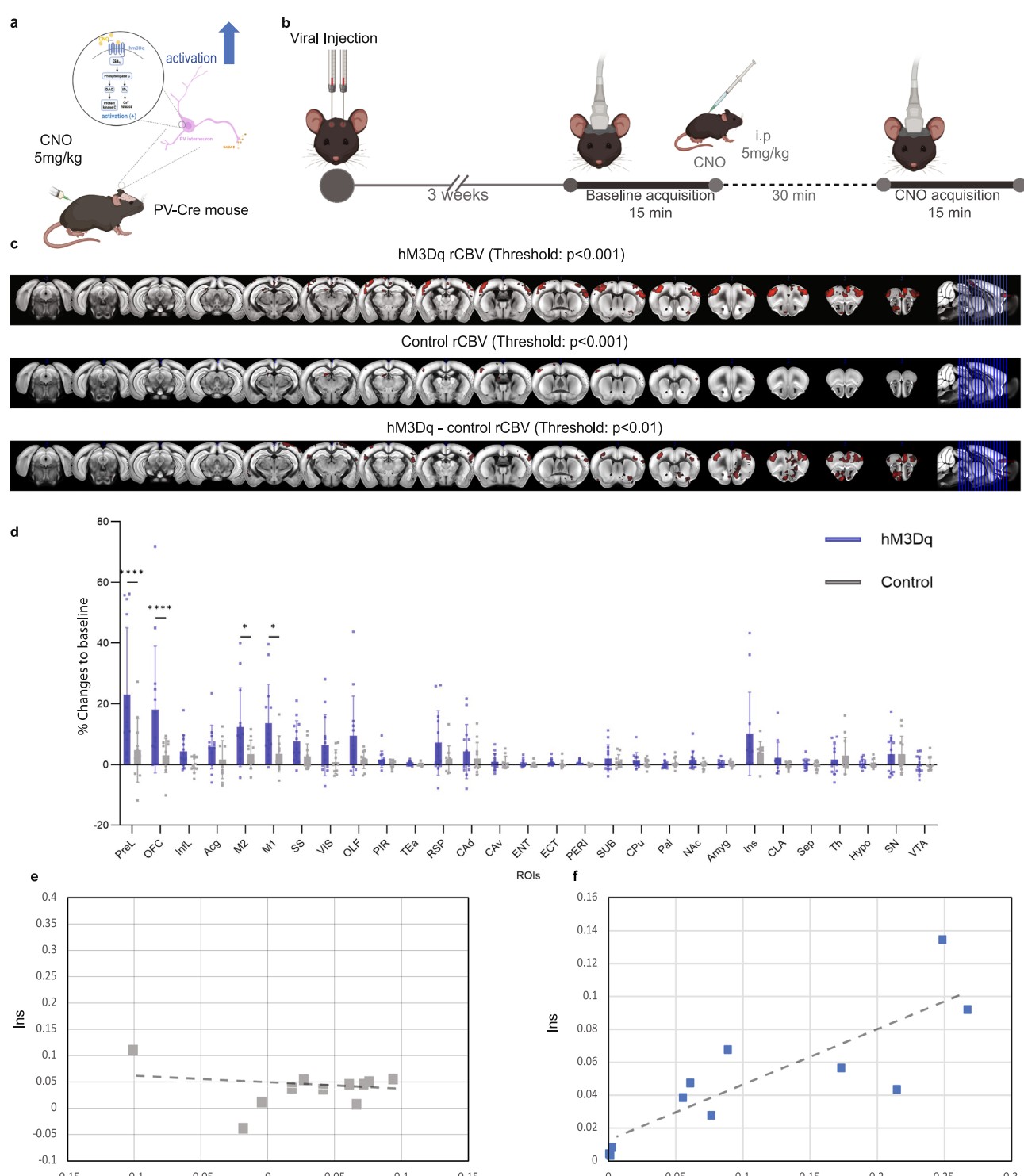

**Fig. 1 | CBV changes following activation of OFC PV interneurons. a** Schematic illustration of PV interneuron activation by hM3Dq DREADD. Activation by clozapine-N-oxide (CNO) leads to increased inhibition of pyramidal neurons, resulting in decreased excitatory output. **b** fUS experimental design. **c** T-score of relative CBV in hM3Dq ($n = 13$) and control ($n = 11$) group along with hM3Dq-control result. **d** rCBV (% changes to the baseline) in 29 ROIs in both hM3Dq and control groups. **e** Correlation coefficient using Pearson's correlation between OFC and Insula in the control group ($r = -0.1882$). The significance of the correlation was tested at a 95% confidence level ($p = 0.5794$). **f** Correlation coefficient using Pearson's correlation between OFC and Insula in the hM3Dq group ($r = 0.8400$).

The significance of the correlation was tested at a 95% confidence level ($p < 0.0001$). PreL prelimbic, OFC orbitofrontal cortex, InfL infralimbic area, Acg anterior cingulate gyrus, M2 secondary motor cortex, M1 primary motor cortex, SS somatosensory area, VIS visual area, AUD auditory area, OLF olfactory area, PIR piriform area, TEa temporal association areas, RSP retrosplenial area, CAd dorsal hippocampus, Cav ventral hippocampus, ventral part, ENT entorhinal area, ECT ectorhinal area, PERI perirhinal area, SUB subiculum, CPu caudate putamen, Pal pallidum, NAc nucleus accumbens, Amyg amygdala, Ins insular area, CLA claustrum, Sep septum, Th thalamus, Hypo hypothalamus, SN substantia nigra, VTA ventral tegmental area, PAG periaqueductal gray, SC superior colliculus.

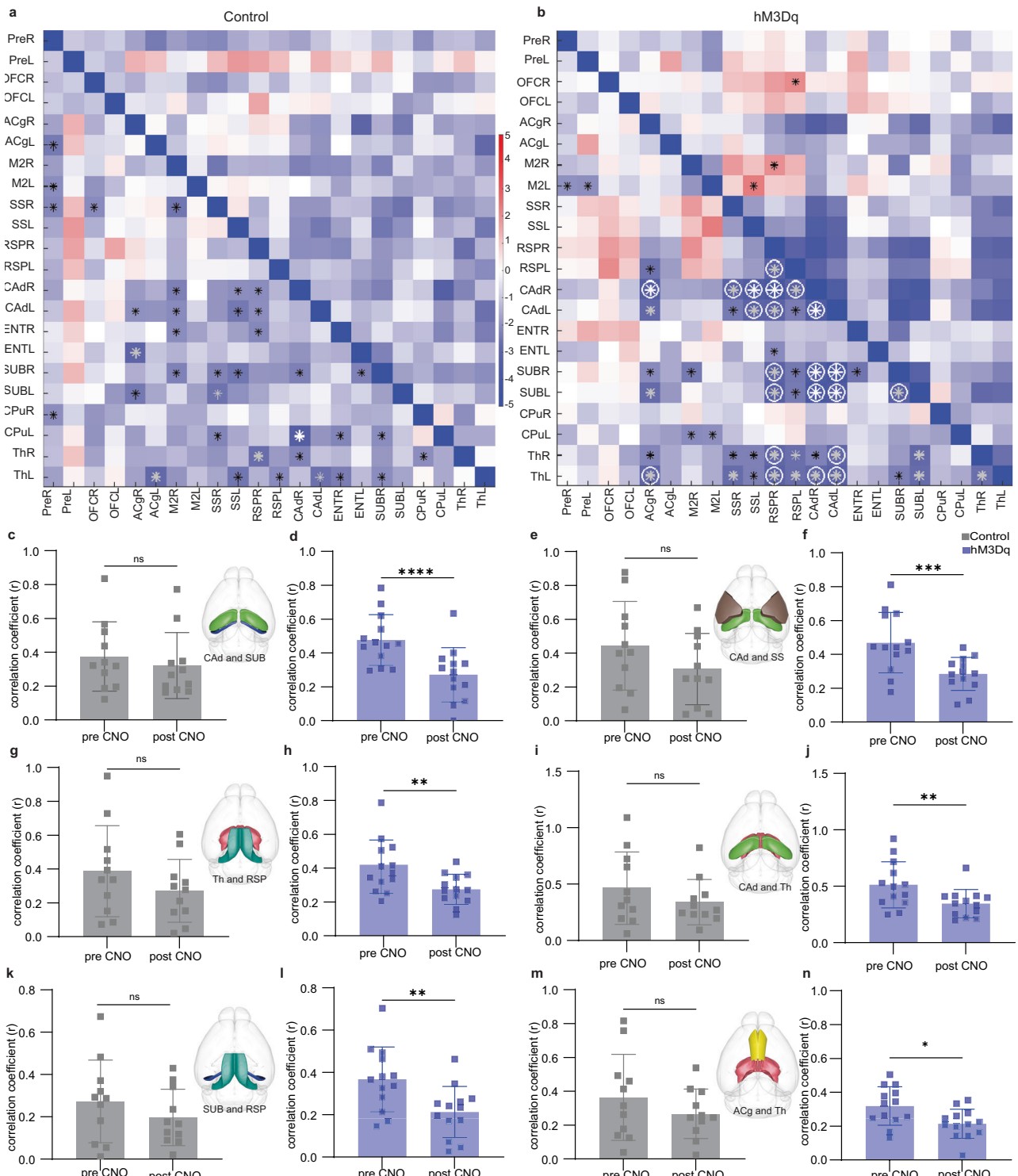

**Fig. 2 | FC alterations in DMN regions following activation of OFC PV interneurons. a, b** The FC matrices show *Z*-scores calculated between the readouts at baseline and post CNO application in 12 different regions (left and right hemisphere) in control and hM3Dq group (paired *T* test, black *$p < 0.05$, gray *$p < 0.01$, white *$p < 0.001$, circle = FDR-corrected). **c, d** Control ($n = 11$) and hM3Dq group ($n = 13$) correlation coefficient (*r*) of FC between dorsal hippocampus (CAd) and subiculum (SUB) in pre CNO (baseline) and post CNO acquisitions (paired *t* test, ****$p < 0.0001$). **e, f** Correlation coefficient (*r*) of FC between somatosensory area

(SS) and CAd (paired *t* test, ***$p < 0.001$). **g, h** Correlation coefficient (*r*) of FC between thalamus (Th) and retrosplenial area (RSP) (paired *t* test, **$p < 0.01$). **i, j** Correlation coefficient (*r*) of FC between CAd and Th (paired *t* test, **p < 0.01). **k, l**, Correlation coefficient (r) of FC between SUB and RSP (paired *t* test, **$p < 0.01$). **m, n**, Correlation coefficient (*r*) of FC between anterior cingulate (ACg) and Th (paired *t* test, *$p < 0.05$). 3D visualization of fUS-scanned regions (Scalable Brain Atlas). Bar graphs represent the mean with error bars indicating SD.

thalamic connections (with RSP and ACg), and the SUB-RSP connection emerged, whose strength was significantly decreased by activation of OFC PV interneurons, with no changes in the control group (Fig. 2c–n). Hemisphere-specific analysis revealed particularly pronounced FC reductions between the right CAd (CAdR) and ipsilateral ACg (ACgR) ($Z = -3.4$, $d = -1.1$) and between the right RSP (RSPR) and both left (ThL; $Z = -3.1$, $d = -0.9$) and right thalamus (ThR; $Z = -2.9$, $d = -0.9$), whereas controls showed no significant changes ($Z < -2.6$, $d > -0.5$) (Fig. 2a, b). Against the general trend of decreasing FC, the hM3Dq group also showed selective CNO-induced FC increases, e.g., between the right OFC and contralateral RSP as well as between left M2 and ipsilateral SS, not seen in controls (Fig. 2a, b). Hemisphere-specific analysis was performed in our study to account for known hemispheric asymmetries and functional lateralization in brain organization. Functional lateralization refers to the preferential engagement of one hemisphere in specific cognitive or neural processes, a phenomenon well-documented in both humans and rodents. Additionally, hemispheric asymmetries in mice, particularly in cortical and subcortical connectivity, have been increasingly recognized, supporting the relevance of lateralized analyses in preclinical models[17–19]. Beyond biological factors, technical considerations may also contribute to apparent lateralization. For example, if signal quality is compromised on one side due to suboptimal gel coupling or increased air content in the skull, this can lead to systematic differences in connectivity at the group level. Including hemisphere-specific analyses helps ensure that both biological and technical sources of asymmetry are appropriately accounted for.

Another point is a trend toward reduced FC observed in the control group following CNO administration, raising the possibility that CNO itself, independent of DREADD expression, may contribute to altered connectivity. Supporting this interpretation, a recent study by Mantas et al. demonstrated that clozapine, the active metabolite of CNO, can affect brain-wide function, mediating changes in sensorimotor gating and connectivity patterns. In that study, functional ultrasound experiments demonstrated that clozapine administration led to a reduction in FC even in wild-type mice. These findings are consistent with our current observations and suggest that the FC decrease in the control group may reflect off-target effects of CNO-derived clozapine, potentially involving receptor interactions such as 5-HT2A, D2, and M1[20].

To investigate if such changes in OFC function and DMN connectivity impair social behavior, as predicted by clinical PRISM studies, we introduced mice in familiar groups of four into the social arena (Fig. 3a), which provides a temporally stable environment to assess naturalistic sociability and exploratory behavior by automated, combined RFID/video-tracking[21]. Our behavioral test offers the capability to monitor and analyze both social and non-social behaviors over extended periods. Commonly used approaches depend on interactions between a manipulated and a non-manipulated animal to isolate specific social responses. While these paradigms are valuable for examining individual-level behavioral changes, our study was deliberately designed to address a complementary and increasingly pertinent question: how neuronal manipulation influences emergent group-level social dynamics within a more naturalistic and ethologically valid context. After 3 days of habituation, two well-established social behaviours, social approach and social sniffing, were scored on two consecutive test days, on which either vehicle or CNO was injected (Fig. 3a). CNO administration significantly reduced social approach and social sniffing durations selectively in the hM3Dq group, during the first 6 h of the dark phase (Fig. 3b, c and Tables S1, S2), and across the whole 12 h dark phase (Fig. S11). Controls showed no difference between CNO and vehicle days (Fig. S11) across the dark phase. To investigate if this reflected a specific social impairment or an unspecific reduction of exploratory behavior, we conducted a novel object recognition test (NORT) after CNO application (Fig. S14). hM3Dq mice showed similar object exploration time as controls (Fig. 3d), suggesting that the observed decrease in social interaction represents a specific social impairment. Importantly, only control mice explored the novel object significantly more than the familiar one, whereas novelty-preference—indicating object-memory—was absent in hM3Dq mice

(Fig. 3d, e). We analyzed locomotion data as a representative non-social behavioral measure in both NORT and social arena experiments. Our results show no significant differences in general locomotor activity between groups, suggesting that the observed changes in social behavior are not due to global behavioral impairments nor motor deficits (Fig. S13). This implies that activating PV interneurons in the OFC causes both social and cognitive symptoms, as seen in patients with schizophrenia and AD.

This study was designed to test the hypothesis of a causal link between OFC activity on the one hand and DMN dysconnectivity and social impairment on the other hand, which was informed by the main clinical findings[2,3] from the PRISM project. The DMN, involving multiple regions including the ventromedial prefrontal cortex (VMPFC) and hippocampal formation (HF), is evolutionarily conserved in rodents, sharing several functional and structural features with humans[22,23]. The rodent DMN equally shows high activity during rest and reduced activity during tasks, indicating similar roles in maintaining baseline brain functions and internal cognitive processes. Our preclinical data highlight significant alterations in hippocampal connectivity, which may contribute to the broader DMN dysconnectivity and could underlie cognitive and social deficits. Our preclinical study supports this idea by showing that decreased FC in DMN regions, particularly in the hippocampus, is connected to social impairment. The observed CBV increases in the PreL, M1, and M2 following chemogenetic activation of PV interneurons in the OFC likely reflect enhanced functional connectivity and network-level modulation. PV interneurons are known to orchestrate cortical network synchrony, particularly through gamma oscillations, which facilitate long-range communication between brain regions[24,25]. Their activation sharpens local excitatory output via feedforward and feedback inhibition, increasing the signal-to-noise ratio and potentially enhancing downstream activation in anatomically connected areas[26]. Given the established anatomical and functional connectivity between the OFC and medial prefrontal as well as motor cortices[27], the increased CBV in PreL, M1, and M2 may reflect a coordinated network response to PV-mediated modulation of OFC output. It is important to note that CBV responses to PV interneuron activation are influenced by several factors, including timing of measurement, brain state, and activation method. In our study, CBV was measured 30 min post-activation, a time point at which vasodilatory effects may dominate. Previous studies have reported similar findings: Vo et al.[29] demonstrated a biphasic vascular response to PV activation, and Anenberg et al.[28] showed increased cerebral blood flow following GABAergic modulation. Additionally, Mantas et al.[20] reported a robust CBV increase after clozapine administration, the active metabolite of CNO[20,28,29].

The observed reductions in FC between DMN regions following chemogenetic activation of OFC PV interneurons likely reflect changes in local circuit dynamics that propagate through large-scale brain networks. PV interneurons are fast-spiking GABAergic cells that exert strong perisomatic inhibition on pyramidal neurons, thereby regulating cortical output. While chemogenetic activation of these interneurons in the OFC is expected to influence local circuit dynamics, the precise impact on overall OFC activity remains to be fully determined. Nevertheless, such modulation may alter the excitatory drive to downstream and functionally connected regions such as the hippocampus, thalamus, retrosplenial cortex, and anterior cingulate cortex. This mechanism is consistent with prior findings showing that PV interneuron activity can modulate long-range network synchrony and memory-related processes by gating pyramidal neuron output[30]. Moreover, the OFC plays a critical role in top-down modulation of limbic and associative cortices. Disruption of this regulatory influence through enhanced local inhibition may desynchronize activity across the DMN, leading to reduced temporal coherence and FC. Similar network-level effects have been observed in studies where PV interneuron activation in prefrontal regions altered fear memory expression and disrupted connectivity with subcortical targets[31,32]. These findings suggest that PV interneuron-mediated inhibition not only shapes local circuit dynamics but also exerts widespread influence on distributed cognitive networks.

**Fig. 3 | Social impairment and memory dysregulation induced by OFC PV interneuron activation.** **a** Behavior experimental design. **b** Social approach duration (average per hour) of individual animals in the control group ($n = 16$) and in the hM3Dq group ($n = 20$) during the first 6 h of the dark cycle after NaCl vehicle versus CNO application (unpaired $T$ test, ***$p < 0.001$). **c** Social sniff duration of individual animals in the control group ($n = 16$) and in the hM3Dq group ($n = 20$) during the first 6 h of dark cycle after NaCl versus CNO application (unpaired $T$ test, **$p < 0.01$). **d** Exploration time of familiar versus novel objects in control (paired $T$ test, *$p < 0.05$) and hM3Dq group. **e** Discrimination index = (time exploring the new object – time exploring the familiar object)/total exploration time in control versus hM3Dq group (unpaired $T$ test, **$p < 0.01$).

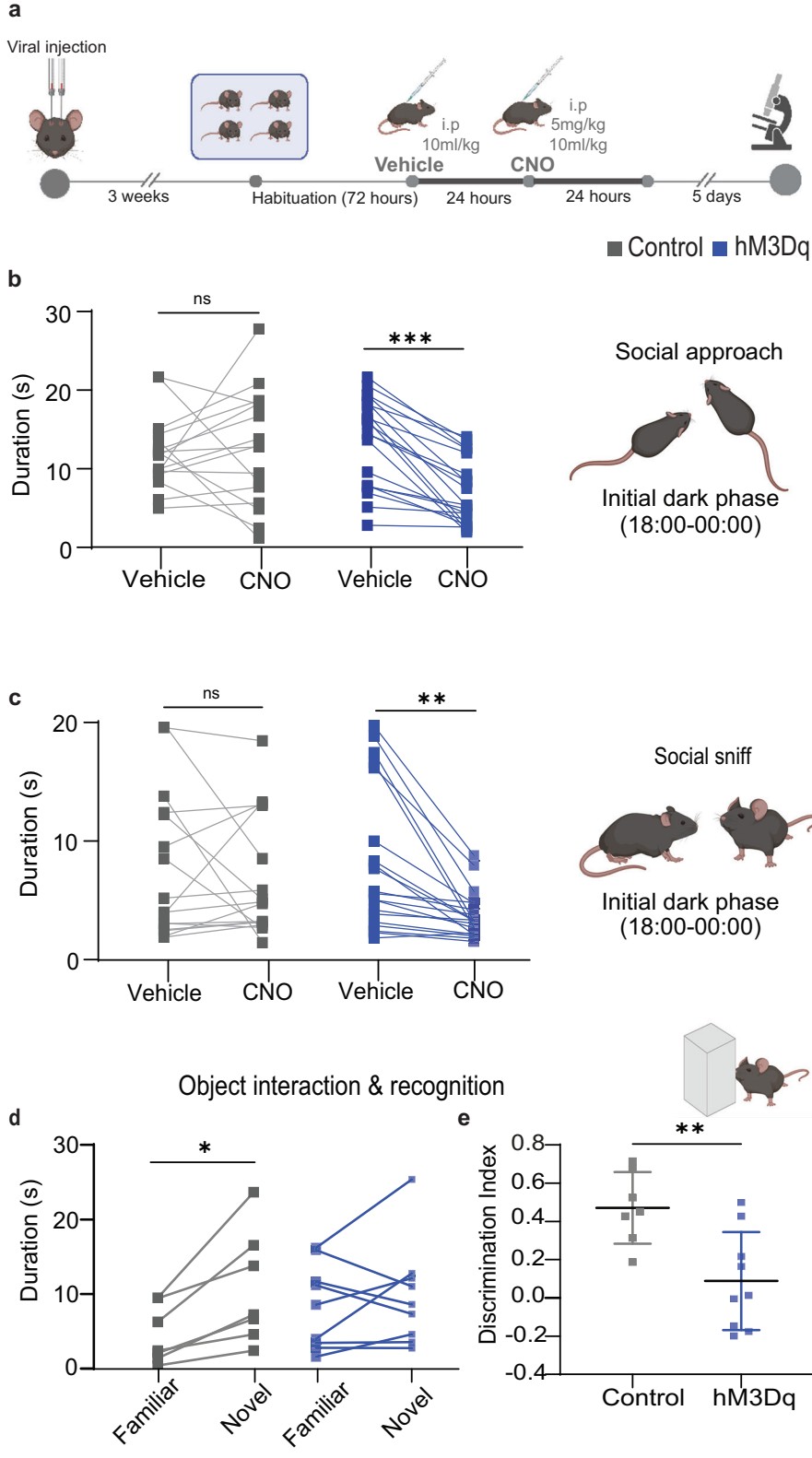

Importantly, such alterations in DMN connectivity mirror patterns observed in neuropsychiatric disorders characterized by PV interneuron dysfunction, including schizophrenia and depression, where reduced FC between the prefrontal cortex and hippocampal-thalamic circuits is a hallmark. Thus, our results provide mechanistic insight into how targeted modulation of inhibitory microcircuits in the OFC can reshape large-scale brain networks.

Although the dominant network-level effect of OFC PV-interneuron activation was a reduction in functional connectivity across canonical DMN nodes, the pattern was not uniformly suppressive; a subset of region pairs

showed increased FC. These increases may result from compensatory recruitment of alternative circuits when DMN-mediated integration is weakened or disinhibition of downstream or parallel pathways caused by altered local balance in the OFC. Such reconfiguration is consistent with the OFC's role as a top-down regulator. These increases are best viewed not as random noise but as a systematic reconfiguration: when OFC PV-interneuron activation desynchronizes DMN hubs, the brain can both (A) recruit alternative sensory- and limbic-driven pathways (compensation) and (B) disinhibit circuits that are normally held in check by OFC output, producing localized hyperconnectivity. Whether such increases are adaptive (restoring function) or maladaptive (producing aberrant synchrony) will depend on behavioral outcome measures and whether the same increases are sustained or transient. Functionally, one would predict that increases involving limbic-sensory axes (e.g., amygdala→sensory cortex, piriform→perirhinal) would bias processing toward externally salient or emotionally laden stimuli, whereas increases involving motor and parietal nodes (e.g., M2 ↔ RSP/SS) would favor sensorimotor re-engagement

Our behavioral setup, as well as other group-based behavioral setups such as Shemesh et al., involves four mice interacting freely within a semi-naturalistic arena. In such environments, mice exhibit interdependent spatial and behavioral patterns that cannot be attributed to individual preferences alone, a phenomenon known as behavioral convergence. These emergent group-level dynamics are highly sensitive to both social composition and environmental context[21,33–39]. Social behavior in group-living species like mice is inherently complex and context-dependent. Traditional paradigms, while informative, may not capture the higher-order interactions and emergent properties that arise in group settings. Our approach, housing either all hM3Dq or all control mice together, was specifically chosen to examine how manipulation influences collective behavior while minimizing potential confounds such as behavior adaptation, which has been observed in mixed groups of animals[40,41]. Behaviorally, chemogenetic PV interneuron activation in the OFC reduced social interaction. The OFC integrates reward/punishment evaluation, emotional regulation, and regulates social behavior via its connections to the amygdala and prefrontal regions[42,43]. OFC circuits modulate socially influenced feeding, rather than eating in general, underscoring its role in complex behavior[9]. Our findings build on this by highlighting the role of GABAergic neurotransmission in the OFC in regulating social behavior, suggesting it as a contributing factor in neuropsychiatric conditions.

We also observed memory impairment, which may be linked to altered FC within the hippocampus and its associated networks. Specifically, the observed reductions in FC between DMN regions likely reflect disrupted integration of cognitive and affective processes. Enhanced inhibition in the OFC may have mechanistically desynchronized pyramidal output, thereby disrupting long-range coordination across these mnemonic circuits[24], though the underlying neuronal mechanisms cannot be determined from the present data. The dorsal hippocampus and subiculum are central to episodic memory and spatial navigation, and their reduced connectivity with the thalamus and somatosensory areas may impair hippocampal-cortical communication essential for contextual processing. Chemogenetic activation of OFC PV interneurons disrupted connectivity within hippocampal–thalamic–retrosplenial–cingulate loops that are central to episodic and contextual memory. Reduced FC between the dorsal hippocampus and subiculum, thalamus, and somatosensory areas likely impaired hippocampal–cortical communication essential for contextual integration[44,45]. The thalamus, acting as a relay hub and a critical relay for aligning hippocampal activity with cortical representations, supports attentional modulation and consciousness, and its weakened connections with the RSP and ACg (two DMN hubs) suggest impaired sensory-cognitive integration[46,47]. The human retrosplenial cortex, involved in auto-biographical memory and scene construction, and the anterior cingulate cortex, implicated in emotional regulation and cognitive control, both showed reduced connectivity in our study that may underlie deficits in internally directed thought and top-down regulation[48,49]. Disruptions in DMN connectivity, particularly involving the hippocampus, have been

associated with cognitive deficits, including memory impairment[50]. Our findings suggest an association between local inhibitory modulation in the OFC and large-scale network alterations involving hippocampal-dependent memory, while not establishing a direct causal relationship[51]. These results are consistent with clinical reports linking social impairment and cognitive changes, but do not allow conclusions regarding the underlying mechanisms of this association.[52–54]

In conclusion, this study demonstrates that chemogenetic activation of OFC PV interneurons disrupts DMN connectivity and results in social and cognitive impairments. These findings advance our understanding of the neurobiological mechanisms underlying social dysfunction and highlight PV interneurons as potentially relevant for elucidating pathways that may be explored in future therapeutic research. Further investigation of PV interneuron modulation in the OFC may provide insights with translational relevance, although its therapeutic relevance remains to be established.

## Limitations
Several limitations of this study should be acknowledged. First, all experiments were conducted exclusively in male mice, restricting generalizability across sexes. Given well-documented sex differences in brain anatomy, physiology, and behavior, future work should include females to assess sex-specific effects.

Second, although we used a sedation protocol combining isoflurane and dexmedetomidine—considered optimal for preserving neurovascular coupling—the influences of anesthesia on FC cannot be entirely excluded. While awake imaging could address this, it introduces stress-related confounds that may themselves alter neural activity.

Third, FC and behavior were measured in separate cohorts, preventing within-subject correlations that could directly link connectivity changes to individual outcomes. Our behavioral assays also focused primarily on sociability and recognition memory, leaving other domains such as anxiety-like behavior, reward sensitivity, and cognitive flexibility unexplored. Fourth, although DREADDs enable targeted manipulation, systemic CNO can convert to clozapine and influence endogenous receptors. Use of next-generation ligands could reduce this confound.

Finally, FC as measured by fUS reflects temporal correlations rather than direct synaptic interactions, limiting causal interpretation of increased or decreased connectivity. Moreover, while our viral strategy targeted OFC PV interneurons, some spread to adjacent regions cannot be ruled out, and hemispheric lateralization was not systematically examined. These considerations highlight the need for future studies combining improved chemogenetic ligands, multimodal readouts (e.g., electrophysiology), and integrated experimental designs to more robustly link local inhibitory modulation in OFC to network-level and behavioral outcomes.

## Methods and material
### Mice
Two cohorts of 6-week-old B6;129P2-Pvalb < tm1(cre)Arbr > /J male mice ($N = 61$; weight range: 18–22 g) were obtained from Charles Rivers Research Models and Services (Germany). Mice were kept under standard conditions (12 h/12 h dark/light cycle, temperature 20–24 °C, humidity 45–65%, and ad libitum access to food and water) in groups of four. After a 2-week acclimation period, stereotaxic surgery was performed. In cohort one of 36 mice, behavioral experiments with or without chemogenetic manipulation were conducted starting 3 weeks after surgery. In cohort two (25 mice), the fUS study was performed. All experiments were approved by the responsible governmental animal ethics committee (Regierungspräsidium Tübingen, Baden-Württemberg, Germany).

### Stereotaxic surgeries
Adeno-associated viral (AAV) vectors were utilized to express either the excitatory DREADD hM3Dq-mCherry or only mCherry (control) in PV+ interneurons of OFC (Fig. S1a). Virus aliquots were stored at −20 °C and thawed prior to the surgery. To prepare the animals for the surgery, 10 mg/ kg of meloxicam (Metacam®, Boehringer Ingelheim, DE) was injected i.p.

20 min before mice were anesthetized with 4% isoflurane in oxygen. Their scalps were shaved under anesthesia before their transfer to a stereotaxic frame (David Kopf Instruments, USA) with 1.5% isoflurane for maintenance of anesthesia. To prevent dehydration, 0.5 ml of NaCl was injected subcutaneously. Throughout the procedure, blood oxygen saturation, temperature, pulse, and respiratory rate were continuously monitored using a biometric paw-attached modular system. A homeothermic warming pad ensured the temperature remained at 37 °C, as measured by a rectal thermometer (PhysioSuite, Kent Scientific Corporation, USA). Bilateral injections of 200 nl viral suspension were made at AP + 2.68 mm, ML ± 0.8 mm, and DV 2.2 mm, relative to bregma, to target OFC[55] using an infusion speed of 50 nl/min, and the needle was withdrawn gradually over 10 min (Figs. S4 and S5). Subsequently, an RFID transponder (1.25 ×8 mm) was subcutaneously implanted near the hindlimb at the back of the mouse.

## Behavioral experiments

Mice were kept in the same group of four subjects throughout the experiment, whereby the four mice had either all received hM3Dq or the mCherry-control vector. Cohort one, 20 hM3Dq mice (5 cages) and 16 control mice (4 cages), participated in the experiment. Starting 3 weeks after surgery, such groups of four mice were housed in a social arena, which provided a temporally stable and more naturalistic social and spatial environment to assess sociability by monitoring the movement of each mouse[21,35]. Each arena provided unlimited access to food and water and contained three nests. The standard dark/light cycle was maintained with 12 h of darkness (6 p.m. to 6 a.m.). Body weight was recorded at transfer to the social arena, before each CNO or vehicle injection, and upon completion of the experiment. After a 3-day habituation phase in the arena (including 1 day baseline, Fig. S12), during which mice were monitored. The duration of the habituation period in our study was informed by previous research demonstrating that both motor activity and social behaviors tend to decline during the initial days following introduction to a new environment. Studies such as Peleh et al. and Kas et al.[21,35,56] have shown that mice exhibit elevated exploratory and social behaviors immediately after placement in a novel arena, which gradually stabilize over approximately 2–3 days. This habituation period is therefore critical to minimize novelty-induced behavioral artifacts and ensure that subsequent measurements reflect baseline, context-adapted behavior. On the subsequent day, the first vehicle injection was conducted. Saline vehicles were injected intraperitoneally (10 ml/kg injection volume) at the beginning of the dark phase. The CNO injection (5 mg/kg, 10 ml/kg injection volume) was conducted the following day, also at the beginning of the dark phase. The selected dose was based on pharmacokinetic and pharmacodynamic data presented in the study by Jendryka et al.[57], which systematically compared the actions of clozapine-N-oxide, clozapine, and compound 21 in DREADD-based chemogenetics in mice. This dose is consistent with previous studies activating excitatory hM3Dq receptors in vivo[58,59].

Their findings indicate that CNO exhibits a relatively low brain penetrance and requires higher systemic doses to achieve effective DREADD activation, particularly in acute experimental designs. In our study, CNO was administered as a single acute injection, and the timing of behavioral testing was aligned with its peak activity window. Lower doses were not tested in this experiment, as our chosen concentration falls within the range commonly used in similar behavioral studies and has been shown to reliably activate hM3Dq receptors without inducing off-target effects or seizure-like activity. No seizure-like behaviors were observed during or after administration, and animals were closely monitored throughout the testing period and 1 day after each set of experiments. A washout phase of at least 5 days followed. To control for potential confounding effects related to injection, handling, or removal from the social arena, we included the vehicle condition as a double control. This design allowed us to isolate the specific behavioral effects of chemogenetic activation from general procedural influences. While our primary comparisons focused on Vehicle versus CNO conditions, we also collected Baseline data during the third day of the habituation phase, when animals were already familiar with the testing environment but had not yet received any injections.

## Novel object recognition test (NORT)

The novel object recognition test (NORT) was conducted over 2 days, with a 1-h delay post-training, based on a modified version of the protocol from Lueptow[60]. The protocol was designed with limited habituation sessions, in line with recommendations by Leger et al. A 1-h interval between training and testing was chosen to assess short-term memory, allowing sufficient time for memory consolidation[61,62]. This timing also corresponds with the pharmacokinetic profile of clozapine-N-oxide (CNO), which was administered acutely prior to the training session. Additionally, this design helps reduce potential confounds related to drug metabolism and behavioral variability. Prior to the NORT, the social arena was modified by removing nests, food hoppers, and water bottles to minimize external distractions (Fig. S14). Twenty mice from cohort one were used for the NORT experiment. During both sessions, object exploration was manually scored based on video recordings. Exploration was defined as the mouse directing its nose toward the object at a distance of ≤2 cm and/or actively sniffing or touching the object. Merely passing by the object without investigative behavior was not counted as exploration. Each session lasted 5 min, but if a mouse did not reach at least 20 s of combined exploration time for both objects within that period, the session was extended beyond 5 min until the 20-s criterion was met. If the mouse failed to reach the 20-s minimum within 10 min, the animal was excluded from further analysis, as insufficient exploration time would not allow a reliable assessment of learning or discrimination. The final results involved 9 hM3Dq mice versus 7 control mice. 4 mice were excluded from the analysis as they did not meet the cut-off duration criteria due to a lack of active periods during the experiment. This exclusion was necessary to ensure data quality and interpretability. Including animals with minimal engagement could have introduced variability unrelated to memory performance. Nonetheless, we are aware that the reduced group size may limit statistical power and should be considered when interpreting the results.

## Functional ultrasound (fUS)

After 3 weeks of viral expression, functional ultrasound imaging was conducted in cohort two (14 hM3Dq and 11 control mice). Sedation was induced using 4% isoflurane in oxygen, which was gradually reduced to 1% upon animal fixation in a stereotaxic frame (David Kopf Instruments, USA). To improve image quality, the scalps were shaved. Heart rate, oxygen level, and body temperature were constantly monitored (Fig. S7), and the temperature was maintained at 37 °C using the PhysioSuite homeothermic warming system (Kent Scientific Corporation, US). A subcutaneous injection of 0.05 mg/kg meloxicam (Metacam®, Boehringer Ingelheim, DE) was administered to reduce possible inflammation. For sedation maintenance, a subcutaneous bolus of dexmedetomidine (0.067 mg/kg, Tocris, USA) was applied, followed by a subcutaneous infusion of dexmedetomidine hydrochloride (0.2 mg/kg/h, 5 mL/kg/h flow rate). Isoflurane was gradually reduced to 0.4%. The ultrasonic gel was applied to the scalp, and the multi-array ultrasonic probe[63] was placed 1 mm above the skin. Since the fUS setup is in a different animal facility than the behavior setup, the functional ultrasound study was performed with a second independent cohort of 14 hM3Dq and 11 control mice. First, an angiographic image was created from 30 coronal slices. Slices were separated by 0.2 mm, starting from the midbrain to the frontal cortex. Image acquisition started 10 min after minimizing the isoflurane concentration (between 0 and 0.4%) throughout the imaging session. The combination of isoflurane and dexmedetomidine employed in our study is among the most widely validated approaches for preserving neurovascular coupling and functional connectivity during imaging[64–66]. To further mitigate potential interference, we deliberately used doses at the lower end of the effective range, resulting in a very light level of sedation. Animals remained in a borderline awake state, as evidenced by their responsiveness to external stimuli (e.g., movement in response to loud noises), indicating preserved sensory reactivity and minimal cortical

suppression. Although awake imaging is a viable alternative, it introduces stress-related artifacts due to physical restraint or head-fixation, which can significantly alter brain activity and vascular responses. Moreover, habituation protocols required for awake imaging are time-intensive and may not be feasible for all experimental designs[67]. Thus, our sedation strategy reflects a careful balance between minimizing stress-induced variability and maintaining physiological relevance.

Each acquisition began with a 15-min baseline (pre-CNO) measurement, followed by intraperitoneal CNO injection. After 30 min, fUS-imaging resumed for another 15 min. After completion of the recording session, mice received atipamezole (Alzane, Zoetis, DE) to antagonize the effect of dexmedetomidine. OFC seed-to-voxel functional connectivity in male C57BL/6 mice with the same sedation condition is provided in Fig. S10.

The regional power Doppler signal was measured in 29 brain regions (as detailed in Table S1) across both hemispheres[68]. After data quality control, one hM3Dq mouse was excluded from the analysis due to strong fluctuations in body temperature during a scan. The remaining data set was used to generate the FC matrices.

## Perfusion and immunohistochemistry

The efficacy of CNO in activating hM3Dq receptors was subsequently validated through cFOS immunohistochemistry. Five days after fUS-imaging, mice received another CNO injection, followed by an overdose of pentobarbital sodium 30 min later and perfusion-fixation with 4% PFA/PBS (Fig. S1b). Intact brains were transferred to 4% PFA/PBS for 24 h, followed by 30% sucrose/PBS until saturation. Brains were cut by microtome into 30 μm slices and stored in an anti-freeze solution (30% ethylene glycol, 30% glycerol, and 40% 10x PBS) at −20 °C. For free-floating histology, two sections per animal were transferred to 24-well plates. The sections were blocked in 0.3% Triton X-100, 10% normal goat serum, and 1% BSA (Dianova, DE) in PBS for 2 h at room temperature with constant agitation. Primary antibodies were diluted in 0.3% Triton X-100, 1% normal goat serum, and 1% BSA in PBS (Dianova, DE) before adding to the wells. The sections were incubated overnight at 4 °C with gentle agitation, using rabbit polyclonal anti c-Fos (abcam, GB, #ab190289, 1:1000) and guinea pig anti-parvalbumin (Synaptic Systems, DE, #195004, 1:500) antibodies. The next day, the sections were thoroughly washed for 2 h, with periodic replacement of the washing solution containing 0.3% Triton X-100, 1% normal goat serum, and 1% BSA (Sigma, US). The sections were then exposed to the secondary antibodies, anti-rabbit Alexa Fluor 488 (Invitrogen, #A11070, 1:1000) and goat anti-guinea pig Alexa Fluor 647 (ThermoFisher, US, #A21450, 1:1000), diluted in 0.3% Triton X-100, 1% normal goat serum, and 1% BSA (Dianova, DE) in PBS. The sections were incubated for 2 h at room temperature with gentle agitation and washed five times for 15 min each. They were then transferred to 24-well glass bottom plates (Sensoplate, Greiner, AT), allowed to air dry, and embedded with an aqueous mounting medium containing DAPI (Fluoroshield™ with DAPI, Sigma, US) (Figs. S2, S3).

## Analysis of behavioral data

After collecting the integrated RFID data with video recordings, the software (SocialScan, CleverSys Inc.) analyzed proximity and movement patterns to identify social interactions. Specifically, social behavior was quantified with detector settings calibrated according to parameters shown in the paper by Peleh et al.[35]. Social sniffing was defined as an interaction occurring when the distance between two mice was less than 3.5 cm and sustained for a minimum duration of 0.33 s. Social approach was characterized by a combination of spatial and movement criteria: the approaching mouse (mouse 1) had to move toward the other mouse (mouse 2) from a distance of less than 100 cm, with a movement angle of less than 45°, covering a minimum distance of 7 cm at a velocity exceeding 4 cm/s. These criteria were selected to ensure consistent and objective identification of social interactions across trials[21]. The primary behavior data sets were then processed in 1-h time bins using MATLAB. Each data point represents the average behavior per hour

per mouse. Mean values were calculated across the entire monitoring phase. All statistical analyses were performed using GraphPad Prism.

## Analysis of fUS data

fUS images were saved in Nifti format and pre-processed using SPM12 (Functional Imaging Laboratory, UK), including realignment, co-registration to the Allen mouse brain atlas, and spatial smoothing with a $0.2 \times 0.2 \times 0.2$ mm³ Gaussian kernel. Regional time courses were extracted for analysis. Acute chemogenetically induced CBV changes were assessed at regional and voxel levels. At the regional level, images were normalized to the baseline (final 2 min before CNO application), and relative CBV (rCBV) time courses were generated by normalizing each region to its average value. For voxel-level analysis, SPM12 performed a first-level general linear model using a pseudo-block approach, comparing the last 2 min of post-CNO acquisition (minutes 59–60) to the baseline (minutes 14–15). Statistical parametric maps were created, and group-level effects were assessed by extracting average t-scores. Functional connectivity (FC) analysis involved bandpass filtering regional time courses (0.01–0.2 Hz). ROI-to-ROI FC matrices were calculated using Pearson's r, transformed to Z-scores via Fisher's Z-transformation, and averaged for group-level matrices (baseline: last 12 min; post-CNO: last 12 min) (Fig. S6). Two-sample t-tests compared groups, generating Z-score matrices. FDR correction (Benjamini-Hochberg method) was applied for multiple comparisons[69].

## Statistics and reproducibility

Details of the statistical analyses and sample sizes are described in the respective sections on behavioral and fUS data analysis. Replicates were defined as individual animals subjected to the same experimental conditions. All experiments were performed double-blinded, once per cohort, and no animals were excluded from analysis unless otherwise stated.

## Reporting summary

Further information on research design is available in the Nature Portfolio Reporting Summary linked to this article.

## Data availability

The experimental data that support the findings of this study are available in Figshare. https://doi.org/10.6084/m9.figshare.30278836.

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

## Acknowledgements

This research project, which resulted in this manuscript, was funded by the Innovative Medicines Initiative 2 Joint Undertaking under grant agreement No. 115916. The Joint Undertaking is supported by the European Union's Horizon 2020 research and innovation program and EFPIA. The content of this publication reflects the views of the author(s) only, and neither the IMI 2JU, EFPIA, nor the European Commission is responsible for any use that may be made of the information contained herein. We acknowledge the use of Microsoft Copilot for enhancing the readability and quality of the text. Some figures in this manuscript were created with BioRender.

## Author contributions

B.H. and H.M.M. developed the scientific concept of the project. B.H. and E.K. designed the experiments and wrote the manuscript. E.K., K.K., T.M.I., F.S. and D.K. contributed to the experiments and data analysis.

## Competing interests

E.K., T.M.I., K.K., F.S., H.M.M. and B.H. are employees of Boehringer Ingelheim Pharma AG. D.K. has no conflict of interest.
