## [Transparent Peer Review file · Communications Biology]

Orbitofrontal PV Interneurons Modulate Social Interaction via Default Mode Network Dynamics

Corresponding Author: Dr Bastian Hengerer

This manuscript has been previously submitted at another journal. This document only contains information relating to versions considered at Communications Biology.

Version 0:

Reviewer comments:

Reviewer #1

(Remarks to the Author)

This manuscript by Khatamsaz et al. investigates how chemogenetic activation of PV interneurons in the orbitofrontal cortex (OFC) influences connectivity within the default mode network (DMN) and social behavior in mice. While the authors aim to demonstrate a mechanistic link between E/I imbalance in the OFC and subsequent social deficits, the manuscript only offers incremental insights and is unable to provide convincing evidence of novelty. The conclusions were partly undermined by its methodological limitations, including reliance on imaging on sedated mice, insufficient mechanistic validation, and limited use of behavioral assays. A relatively novel aspect of the study is the use of functional ultrasound, a sort of like a fMRI for rats, hence achieving the spatial resolution of fMRI without the need for an actual fMRI. This makes it very useful in animal studies and helps better correlate animal brain network studies with human studies. This study is hence more of a demonstration that functional ultrasound technology could allow for human functional connectivity results to be replicated in mice. This therefore opens the avenue to explore the neural circuit mechanisms behind network changes that could not be done in humans. Some major and minor concerns are as follows:

Major concerns:

- (1) The use of sedation (isoflurane and dexmedetomidine, as described in lines 262-285) during fUS-imaging could lead to significant alterations in cerebrovascular dynamics as well as neuronal activity. This could introduce artifactual or non-representative measures of cerebral blood volume and compromise the functional connectivity results. Therefore, caution has to be taken when attempting to establish a relationship between changes in DMN connectivity during sedation and subsequent social behavior in awake behaving mice.
- (2) In fact, the study can be regarded as two separate studies because different mice were used in the behavioural and imaging study, and it relies on the already well established literature of the DMN's functional role in social behaviour to tie their two parallel studies together. The authors aim to suggest a causal link between OFC E/I imbalance, DMN dysconnectivity, and social deficits (line 88-89). However, the evidence presented remains correlational and the observed connectivity alterations could be secondary effects rather than the actual drivers of subsequent social impairments. Since it is well established that the orbitofrontal/prefrontal cortex plays a vital role in social behaviour and memory, the behavioral results are well expected.
- (3) The assessment of social behavior relies heavily on the automated RFID-supported video tracking analysis, which only managed to capture tens of seconds of social approach and social sniff throughout the six hours of recording in the dark phase. Could this truly reflect the mice's sociability and impairment in social behavior? More social behavioral tests, such as the three-chamber social test, social recognition test, resident-intruder test, should be included to strengthen the rigor of this finding. Similarly, the conclusion of cognitive deficit induced by PV-activation based solely on the novel object recognition test is not well-supported.
- (4) It would be better for the paper to actually have some network analysis of the DMN network in control mice to show the centrality of the orbitofrontal cortex in order to provide some insight into why manipulations to the orbitofrontal cortex hold

such widespread effects over the entire DMN.

(5) Please clarify why female mice were not included in the study. Including females would greatly enhance the study's relevance, or the authors should at least address this limitation explicitly.

Minor concerns:

(1) Line 20: The abbreviation "SZ" for schizophrenia is not necessary since it's only used once throughout the manuscript.

(2) Fig. 1c-d: Does chemogenetic activation of PV interneuron lead to a brain-wide increase in cerebral blood volume (CBV)? Or were ROIs with reduced CBV excluded from the analysis? This has to be clarified, especially given the expectation that PV activation might primarily reduce neuronal activity and cerebral blood flow (see <https://doi.org/10.1177/0271678X20930831>).

(3) Fig. S1a: Confocal images showing the spread of hM3Dq expression in the OFC should be provided.

(4) Fig. S1c: Confocal images comparing the proportion of Fos-positive cells in AAV-hSyn-DIO-hM3Dq-mCherry-injected subjects and AAV-hSyn-DIO-mCherry-injected control should be provided.

(5) Line 43: The timing of manipulation (i.e., 14 days post-surgery) might be somewhat premature to allow for sufficient expression of the hM3Dq, especially under the Cre-dependent promoter.

(6) Line 47-50: The reported increase in CBV within the olfactory area (OLF) was not supported by the hemispheric analysis (Fig. S2e), which shows statistically insignificant increase in both hemispheres of this region.

Reviewer #2

(Remarks to the Author)

In the manuscript by Khatamsaz et al., the authors investigate the neural mechanisms behind DMN-mediated disruption to social interaction in a mouse model. The authors hypothesized the PV interneuron function in the OFC would modulate E/I balance in the DMN, and that this E/I balance would affect the connectivity in the DMN they thought might be involved in social interaction. To test this hypothesis, the authors utilized hM3Dq-expressing mice and control mice and measured brain activity using functional ultrasound imaging as well as social behavior in the two mouse groups. The authors present an interesting hypothesis, and a notable method in the application of fUS imaging in conjunction with selective modulation of PV interneurons in the OFC.

However, I have two broad criticisms of this work that prevent me from recommending it for publication in its present state:

1. First, the authors' primary neurophysiological endpoint is very broad. Fifteen minutes of widespread changes in functional connectivity across the DMN (without careful statistical accounting for inter-animal variance) is not sufficient evidence that OFC-PV interneuron activation modulates DMN connectivity in a specific and reliable manner. Can the authors demonstrate that alternative interventions (e.g. modified interneuron activity in other brain regions) do not elicit similar broad changes?

The existing control is not sufficient to isolate the level of specificity and selectivity necessary to highlight potential mechanism. This is especially critical given that the primary strength of the authors' use of fUS imaging is the high spatiotemporal resolution afforded by this approach in resolving microcircuit activity.

2. Similarly, I am not convinced by the changes in the primary behavioral endpoints: social approach, social sniff, and object recognition and interaction tests. Is there any reason to believe that modifying interneuron activity in any region of the brain would not disrupt these general behaviors, even without DMN functional connectivity changes? Is there any reason to believe that these disruptions are selective to social behaviors or object memory (e.g. are the mice impaired across a range of non-social behavioral measures?).

Without more extensive intervention controls and behavioral testing, I find it difficult to be convinced that there is a specific, mechanistic role for PV interneurons in social interaction in mice, much less its translational potential for human generalization.

On a more minor note, the theoretical foundation of the study requires further clarification. The authors posit that DMN connectivity is related to E/I balance, but fUS imaging is conducted 30 minutes after CNO injection. The authors should provide further argument that this methodology is actually appropriate for assessing E/I balance as a mechanistic component of DMN connectivity.

Reviewer #3

(Remarks to the Author)

Brief summary of the manuscript

The authors used DREADD based chemogenetics to activate Parvalbumin positive (PV+) neurons in the orbitofrontal cortex (OFC) and measured following large scale connectivity and social as well as memory related behavior in the mouse. Increasing the activity of PV+ neurons in the OFC led according to the authors to reduced connectivity within the default mode network (DMN) as well as reduced social behavior and impaired memory.

Overall impression of the work

The authors aim to address an interesting and potentially clinically important question. Furthermore, the application of novel

techniques, such as functional ultrasound imaging (fUS), adds innovation to the manuscript. While the manuscript presents some potentially valuable observations, the conciseness of the presented data limits the strength and depth of the conclusions. Indeed, some crucial control experiments as well as further clarifications on the methodology and data analysis are missing and needed to support the claims made. Overall, while the research question and approach are compelling, some of the interpretations in the current version appear overstated or insufficiently supported by the available evidence.

Specific comments, with recommendations for addressing each comment

1. The authors perform immunohistochemistry experiments to assess injection site and activation of hM3Dq receptors in OFC PV+ neurons. However, the images shown in the manuscript do not allow for proper assessment of the overall injection spread and comparison with the control group. In order to, evaluate the correct target of the OFC it would be helpful if authors include images at lower resolution that demonstrate correct targeting of the area. Furthermore, adding images for both groups would help to visually compare expression patterns, in addition to the bar graph represented. Additional evaluation of expression in multiple mice and along also a frontal caudal axis via for example frequency mapping would strengthen the manuscript, but is not critical.

2. The authors assess the effectiveness of the chemogenetic activation via cFos staining and claim that the manipulation leads to increased firing of PV+ neurons which in turn causes higher inhibition of pyramidal cells and thereby generally lower excitatory output of the area. However, cFos staining alone does not provide adequate evidence of activation of the targeted neurons nor the effect on the local circuit, to support this claim. In order to, evaluate whether the manipulation produces the intended physiological effect and make definitive claims about increased inhibition and altered circuit output, in vivo electrophysiological experiments are necessary control experiments. Along the same line, the authors refer to E/I imbalance but do not show evidence of the presence of such an imbalance. This is rather problematic, in the absence of direct evidence of an imbalance, these statements are speculative and should be omitted.

3. The CNO dosage used in this study appears relatively high for this type of experiment (e.g., Gomez et al., 2017). It would be helpful if the authors add a justification for the chosen dose, as well as clarifications whether lower doses were tested and whether measurements for seizure like activity were conducted. While a pharmacokinetic analysis may not be necessary, any additional information on dose - response observations, or references to prior work using similar concentrations in comparable contexts, would help to support the validity of this approach.

4. In Figure 1 authors show CBV changes following their manipulation, however, in its current presentation the figure has a few issues that limit its clarity and interpretability. Below are some suggestions that may help the evaluation of this result.

4.1. In Figure 1c there are marked blobs outside of the brain mask, I suppose this might be a problem related to the registration process. I would recommend the authors verify correct registration and adjust the illustration.

4.2. For the interpretation of Figure 1d an illustration with ROI locations as well as single data points, would be helpful.

4.3. The correlation presented in Figure 1f seems to include two outliers. It would be interesting to see the same correlation without these datapoints.

5. In Figure 2 FC alterations are displayed. The figure shows ingroup as well as between group comparisons which is very helpful for the interpretation of the result. However, starting from panel 2c onward, there appears to be a trend towards reduced connectivity also in the control group. The authors should elaborate on the potential effect of CNO in the absence of the DREADD receptor, especially considering the relatively high dosage used. It might also be worth to account for CNO-related effects in the analysis.

6. In Figure S3 authors show FC matrix for each group, however in its current form the representation and statistical approach used are not entirely clear.

If each matrix shows baseline vs. post-CNO comparison within one group, it appears that CNO administration leads to FC alterations in the control group. The authors should elaborate on this finding, especially considering the rather high CNO dosage and its potential off-target effects. In addition, such a within group comparison should be analyzed using paired statistical testing, since it refers to repeated measures within the same animals. Lastly, adding a matrix that demonstrates the between group differences in baseline as well as post-CNO window would enhance the interpretation of these effects. Alternatively, if the matrix shows alterations within group but the significance markers reflect between group comparisons, unpaired statistics are appropriate, but within group statistical tests would be missing. The addition of these comparisons would be beneficial to demonstrate that CNO does not affect FC in control animals and that the baseline connectivity is comparable between groups.

7. Authors mention how activation of OFC PV+ neurons leads to CBV and connectivity changes in various areas of the brain. Further elaboration on the significance and rationale of these areas and the results itself would add value and help contextualize the findings. Below some examples where this could be beneficial.

7.1. Line 46 onwards. Authors mention areas that show CBV increase, providing a rationale for why these specific areas show the effect would aid the interpretation of the result.

7.2. Line 53 onwards. Similarly, for the reported reductions in FC, further elaboration on the specific brain regions and the potential relevance of these alterations would help readers understand the implications of these findings.

7.3. Line 61 onwards. Authors should elaborate on the rationale behind the hemisphere-specific analysis of FC.

7.4. Line 65 onwards. Authors report that against the general trend of decreased FC, some areas also show increased FC. Further elaboration on the relevance of this result would add value to the finding.

8. For fUs experiments authors state that heart rate, and other physiological parameters were monitored, including a supplementary illustration of these measures would enhance interpretation of the results.

9. In order to, measure social behavior authors placed animals in an automated tracking apparatus and measured naturalistic behavior over multiple days. Further explanation of this approach along with references to previous studies using similar methods would help contextualize these results. Specifically, clarifications regarding the following methodological details would facilitate interpretation of the results.

9.1. Social behavior was assessed within groups of either all hM3Dq or all control mice. However, in many established social behavior paradigms, interactions are typically measured between manipulated and non-manipulated control animals. A discussion regarding the chosen experimental design and possible confounding factors should be added.

Moreover, including a second measure of social behavior with a manipulated vs. non manipulated control animal comparison would strengthen the conclusions and help rule out group-level behavioral confounds.

9.2. Adding a justification for the duration of the habituation period with references to previous studies would enhance clarity.

9.3. The method used to score social behavior, including the definition of what constitutes social approach or social sniffing, should be described in greater detail.

9.4. Further elaboration on the rationale behind the vehicle vs. CNO approach (rather than baseline vs. post-CNO) would enhance understanding.

9.5. Further explanation on data representation should be added in order to facilitate comprehension of the results. For example, are single data points individual bouts of behavior or per-animal averages? Clarifying this would help evaluate the variability and consistency of the effect across animals. If possible, displaying individual mouse data could further strengthen the analysis.

10. In order to investigate whether alterations in social behavior were specific to this behavioral domain or rather reflect a reduction in exploratory behavior, the authors performed a NOR task. Further explanations on this task and the resulting effect would give additional value to this finding.

10.1. Further elaboration on the choice of the 1-hour interval between training and testing session would be helpful. Offering a rationale for this duration, as opposed to longer intervals commonly used, would provide valuable context.

10.2. The method used to measure and score object exploration should be described in greater detail.

10.3. The authors mention that 4 mice did not meet the cut off duration due to a lack of active periods. Considering the group sizes (n=9 and n=7), the exclusion of 4 mice seems substantial. Adding a discussion regarding the potential impact of this exclusion on the analysis would enhance clarity.

10.4. Although not necessary, additional controls assessing general locomotor activity (even as measured within the same task) would help rule out confounding effects. Similarly, if the authors would like to discuss cognitive impairments more broadly, including further assessments of other types of memory or other cognitive domains could strengthen and expand the conclusions, though this is not essential and might be outside the scope of this work.

10.5. It would be helpful if the authors could expand the discussion of the observed memory impairments, including possible underlying mechanisms and their relevance to the study's overall conclusions. Additionally, considering how these memory impairments might relate to or influence the reductions in social behavior could provide further valuable insight.

11. The authors describe how altering OFC PV activity affects both connectivity and behavior. In order to, link these two main results, the authors might consider exploring this relationship between connectivity and behavioral alterations using for example multivariate modelling approaches. While this could provide valuable insights, the use of different cohorts for the two measures might make this a difficult endeavor and such an analysis is not essential.

Version 1:

Reviewer comments:

Reviewer #1

(Remarks to the Author)

Thanks for the responses to the questions raised. I have no further comments.

Reviewer #2

(Remarks to the Author)

The authors have reasonably addressed my concerns.

Reviewer #3

(Remarks to the Author)

Brief Summary of the Manuscript

In this study, the authors used DREADD-based chemogenetics to activate parvalbumin-positive (PV⁺) neurons in the orbitofrontal cortex (OFC) and subsequently assessed large-scale functional connectivity as well as social and memory-related behaviors in mice. According to the authors, increasing PV⁺ neuron activity in the OFC led to reduced connectivity within the default mode network (DMN), accompanied by decreases in social behavior and impairments in memory

performance. In the revised manuscript, the authors have added several methodological clarifications and additional control measurements to support their conclusions.

Overall Impression of the Work

I thank the authors for their careful efforts in revising the manuscript and for addressing the reviewers' previous comments. The added clarifications have substantially improved the interpretability and readability of the manuscript, and the inclusion of new control experiments strengthens the work.

However, while the revisions have improved the contextualization and clarity of the manuscript, and the authors have made a clear effort to moderate their interpretations, in several instances the presented data remain too limited to fully support some of the claims. In addition, a few of the previous comments have not been completely addressed. Overall, the revision represents a clear improvement, but some additional refinement would further strengthen the manuscript and ensure that the conclusions are well supported by the data.

Specific comments

1. The addition of lower-resolution confocal images illustrating viral expression is appreciated. However, the overall image quality could still be improved. In Figure S5 (hM3Dq group), the extent and intensity of viral expression appear somewhat variable across animals. It might be helpful to briefly comment on this variability or provide a summary figure showing expression extent across animals.

2. While the authors did not provide the electrophysiological control experiments previously suggested, I appreciate their revised and more cautious interpretation of the c-Fos data, as well as their restraint from making definitive claims regarding altered circuit output or excitation/inhibition balance.

2.1 In this context, the authors should note that the legend of Figure S14 still mentions an "E/I imbalance," which should be revised for consistency.

2.2 Additionally, Figure 1a schematically depicts the effect of PV⁺ neuron activation on local pyramidal neurons. Since the manuscript does not provide experimental data demonstrating how this manipulation affects other neuronal populations, I recommend refraining from detailed circuit-level interpretations in the schematic and corresponding legend.

3. I appreciate the authors' detailed explanation regarding the chosen CNO dosage, and especially the note that animals were carefully monitored and did not exhibit seizure-like behavior. However, the justification based on Jendryka et al. (2019) is not entirely adequate. The study by Jendryka et al. does not report specific data for a 5 mg/kg dose, and although the authors suggest that dose ranges from 3 to 5 mg/kg CNO should be effective, the study explicitly refers to inhibitory DREADD receptors (hM4Di), not excitatory hM3Dq receptors as used here. Jendryka et al. further emphasize that receptor type significantly influences effective dosing. I therefore suggest that the authors reference additional studies supporting the chosen dose for excitatory DREADDs, or provide further clarification.

4. I thank the authors for their added explanation regarding the FC alterations observed in the control group (Figure S9).

4.1 However, the authors did not address my previous comment regarding the use of paired statistical tests for within-group comparisons.

4.2 In addition, including a connectivity matrix that depicts between-group differences both at baseline and post-CNO would substantially enhance interpretability.

4.3 Lastly, in Figure S9 the "circled stars" denoting FDR-corrected results are barely visible; this might be a resolution issue that could be easily improved.

5. The expanded explanation of the social behavioral measurements is very helpful and greatly improves clarity.

5.1 The rationale for the chosen test is now clearly stated. Nevertheless, given that social behavior is inherently reciprocal, it could be informative to complement the current paradigm with a classical social interaction test involving an unaffected stimulus mouse. Although such tests have limitations, their inclusion could provide additional insight into different aspects of social behavior.

5.2 Similarly, while adding further memory-related tests may not be feasible, doing so would strengthen the claims regarding memory impairments.

5.3 Baseline data for social approach were appropriately added in Figure S12; however, equivalent baseline data for social sniffing are still missing and should be included.

5.4 While vehicle versus CNO comparisons are relevant, including analyses comparing the DREADD and control groups under the various conditions would add valuable context. For instance, a mixed linear model could be employed to appropriately test these interactions.

5.5 I appreciate the inclusion of locomotion measures as behavioral controls. However, Figure S13 should display individual data points, and the legend should be corrected to avoid indicating statistical significance where none is present.

5.6 Finally, the explanation for the exclusion of four mice from the NOR test is appreciated, but a brief discussion addressing why several animals did not reach the inclusion threshold would provide useful context, though this is only a minor suggestion.

6. It may be beneficial to further moderate statements implying mechanistic or causal relationships, particularly regarding potential treatment implications. Given the correlational design and use of separate experimental cohorts, more cautious language would align well with the data and maintain a balanced interpretation.

Version 2:

Reviewer comments:

Reviewer #3

(Remarks to the Author)

Thank you, the authors have reasonably addressed my concerns.

Reviewer #1 (Remarks to the Author): Major concerns

Comment	Response
(1) The use of sedation (isoflurane and dexmedetomidine, as described in lines 262-285) during fUS-imaging could lead to significant alterations in cerebrovascular dynamics as well as neuronal activity. This could introduce artifactual or non-representative measures of cerebral blood volume and compromise the functional connectivity results. Therefore, caution has to be taken when attempting to establish a relationship between changes in DMN connectivity during sedation and subsequent social behavior in awake behaving mice.	We appreciate the reviewer's concern regarding the potential confounding effects of sedation on cerebrovascular dynamics and neuronal activity during functional ultrasound (fUS) imaging. Indeed, no sedation protocol is entirely free from physiological impact. However, the combination of isoflurane and dexmedetomidine used in our study is among the most widely recommended and validated approaches in the literature for minimizing disruption to neurovascular coupling and preserving functional connectivity patterns during imaging ¹⁻³. Importantly, the dosage employed in our protocol was deliberately kept at the lower end of the effective range, resulting in a very light level of sedation. Animals remained in a borderline awake state, as evidenced by their responsiveness to external stimuli—such as movement in response to loud noises—indicating preserved sensory reactivity and minimal suppression of cortical activity. While awake imaging is an alternative, it introduces its own set of challenges, particularly stress-related artifacts due to the need for physical restraint or head-fixation. Such stress can significantly alter brain activity and vascular responses, potentially confounding the interpretation of functional connectivity data. Moreover, habituation protocols required for awake imaging are time-consuming and may not be feasible for all experimental designs ⁴. Therefore, our choice of sedation reflects a careful balance between minimizing stress-induced variability and preserving physiological relevance. We acknowledge the limitations and have taken steps to interpret our findings with appropriate caution, especially when relating DMN connectivity changes under sedation to subsequent behavior in awake animals. We have addressed this point in the revised manuscript (lines 258–262, Second, although we used a sedation protocol combining isoflurane and dexmedetomidine—considered optimal for preserving neurovascular coupling— influences of anesthesia on FC cannot be entirely excluded. While awake imaging could

	address this, it introduces stress-related confounds that may themselves alter neural activity).
(2) In fact, the study can be regarded as two separate studies because different mice were used in the behavioural and imaging study, and it relies on the already well established literature of the DMN's functional role in social behaviour to tie their two parallel studies together. The authors aim to suggest a causal link between OFC E/I imbalance, DMN dysconnectivity, and social deficits (line 88-89). However, the evidence presented remains correlational and the observed connectivity alterations could be secondary effects rather than the actual drivers of subsequent social impairments. Since it is well established that the orbitofrontal/prefrontal cortex plays a vital role in social behaviour and memory, the behavioral results are well expected.	Point 2 is addressed through responses to several other reviewer comments, which elaborate on the relationship between the behavioral and imaging findings and clarify the interpretational framework. The manuscript has been modified accordingly.
(3) The assessment of social behavior relies heavily on the automated RFID-supported video tracking analysis, which only managed to capture tens of seconds of social approach and social sniff throughout the six hours of recording in the dark phase. Could this truly reflect the mice's sociability and impairment in social behavior? More social behavioral tests, such as the three-chamber social test, social recognition test, resident-intruder test, should be included to strengthen the rigor of this finding. Similarly, the conclusion of cognitive deficit induced by PV-activation based solely on the novel object recognition test is not well-supported.	While the absolute time spent in social behaviors such as social approach and sniffing may appear brief, it is important to note that these data are calculated in each mouse as averages per hour across the entire six-hour dark phase. This approach allows us to capture consistent patterns of social engagement over time, rather than relying on isolated events. Moreover, we have clarified this point in the revised manuscript (see Methods and Results sections, lines 513-515, Each data point represents the average behaviour per hour per mouse. Mean values were calculated across the entire monitoring phase), emphasizing that the reported durations reflect hourly averages and not cumulative totals. To answer the point regarding other social tests, much of what we currently understand about mouse social behavior stems from short-term experiments involving isolated pairs of animals. These tightly controlled dyadic setups are relatively straightforward to implement in laboratory settings and have yielded valuable insights across numerous studies. However, in natural environments, mice typically live in groups characterized by dynamic and intricate social hierarchies that shift based on ecological factors like habitat and resource availability. As a

	result, behaviors observed in artificial or simplified social contexts may not accurately reflect natural social phenotypes and could be misinterpreted or overlooked entirely.
(4) It would be better for the paper to actually have some network analysis of the DMN network in control mice to show the centrality of the orbitofrontal cortex in order to provide some insight into why manipulations to the orbitofrontal cortex hold such widespread effects over the entire DMN.	Resting-state imaging studies in rodents consistently identify the OFC as a central node within intrinsic connectivity networks. In mice, BOLD and CBV-weighted fMRI revealed strong functional connectivity between the OFC and regions such as the retrosplenial cortex, ventral hippocampus, thalamus, and prefrontal cortex, forming a network analogous to DMN. In rats, similar findings under sedation showed the OFC—particularly its ventral, lateral, and rostral medial subdivisions—integrated within a DMN-like circuit that includes the prelimbic cortex, cingulate cortex, posterior parietal cortex, and hippocampus^{5,6}. We have addressed this point in the revised manuscript (lines 33–38, In mice, the OFC shows strong functional connectivity with regions such as the retrosplenial cortex, ventral hippocampus, thalamus, and prefrontal cortex and in rats, the subdivisions of the OFC have similarly been shown to integrate within a DMN-like circuit involving the prelimbic cortex, cingulate cortex, posterior parietal cortex, and hippocampus). Supporting these observations, we provide new data from a large-scale functional ultrasound imaging dataset collected in 44 male C57BL/6 mice (aged 7–9 weeks) under matching sedation conditions. This dataset shows robust OFC seed-to-voxel connectivity, with statistical analyses confirming strong associations with DMN regions. These results, illustrated in Fig. S10, demonstrate the centrality of the OFC within the DMN and provide insight into why manipulations to this region exert widespread effects across the network.
(5) Please clarify why female mice were not included in the study. Including females would greatly enhance the study’s relevance, or the authors should at least address this limitation explicitly.	We have addressed this point in the revised manuscript, limitation section (lines 254–257, Several limitations of this study should be acknowledged. First, all experiments were conducted exclusively in male mice, restricting generalizability across sexes. Given well-documented sex differences in brain anatomy,

	physiology, and behaviour, future work should include females to assess sex-specific effects).
Minor concerns	
(1) Line 20: The abbreviation “SZ” for schizophrenia is not necessary since it’s only used once throughout the manuscript.	Corrected.
(2) Fig. 1c-d: Does chemogenetic activation of PV interneuron lead to a brain-wide increase in cerebral blood volume (CBV)? Or were ROIs with reduced CBV excluded from the analysis? This has to be clarified, especially given the expectation that PV activation might primarily reduce neuronal activity and cerebral blood flow (see https://doi.org/10.1177/0271678X20930831).	All recorded ROIs were included in the analysis. In our study, nearly all ROIs showed an increase in CBV after CNO application relative to baseline in both groups, while the hM3Dq group exhibiting a significantly greater increase. Several points should be considered when interpreting this data: First, Mantas et al. (2025) (doi.org/10.1038/s41386-025-02060-z) reported a robust CBV increase following administration of clozapine (the active metabolite of CNO), as shown in their Supplementary Figure 5. This supports the notion that Clozapine and CNO can increase CBV ⁷. Second, the effect of PV interneurons activation on CBV is known to depend on multiple factors, including the timing of measurement (early vs. late response), brain state (awake vs. anesthetized), and the method of activation etc. For example, Vo et al. (2023) (doi.org/10.1073/pnas.2220777120) demonstrated that PV interneuron activation induces a biphasic vascular response: an initial fast vasoconstriction followed by a slower vasodilation mediated by substance P. This underscores the importance of measurement timing as in our study CBV values were measured 30 minutes after interneurons activation⁸. Finally, consistent with our findings, Anenberg et al. reported that optogenetic modulation of GABAergic neurons leads to increased cerebral blood flow, further supporting the observed CBV changes ⁹ (Lines 148-155, It is important to note that CBV responses to PV interneuron activation are influenced by several factors, including timing of measurement, brain state, and activation method. In our study, CBV was measured 30 minutes post-activation, a time point at which vasodilatory effects may

	dominate. Previous studies have reported similar findings: Vo et al. (2023) demonstrated a biphasic vascular response to PV activation, and Anenberg et al. (2015) showed increased cerebral blood flow following GABAergic modulation. Additionally, Mantas et al. (2025) reported a robust CBV increase after clozapine administration, the active metabolite of CNO).
(3) Fig. S1a: Confocal images showing the spread of hM3Dq expression in the OFC should be provided.	Confocal images with lower resolution demonstrating the injection sites for both groups are provided in Fig. S4-S5.
(4) Fig. S1c: Confocal images comparing the proportion of Fos-positive cells in AAV-hSyn-DIO-hM3Dq-mCherry-injected subjects and AAV-hSyn-DIO-mCherry-injected control should be provided.	Confocal images comparing the proportion of Fos-positive cells are provided in Fig. S2 (control group) and Fig. S3 (hM3Dq group), respectively.
(5) Line 43: The timing of manipulation (i.e., 14 days post-surgery) might be somewhat premature to allow for sufficient expression of the hM3Dq, especially under the Cre-dependent promoter.	We appreciate the reviewer’s observation regarding the timing of the manipulation. The fUS experiment was conducted three weeks post-surgery (similar to behavioral experiment), allowing sufficient time for robust expression of hM3Dq under the Cre-dependent promoter. The mention of “14 days” in line 43 was a typographical error and has been corrected in the revised manuscript.
(6) Line 47-50: The reported increase in CBV within the olfactory area (OLF) was not supported by the hemispheric analysis (Fig. S2e), which shows statistically insignificant increase in both hemispheres of this region.	We thank the reviewer for this insightful observation. The discrepancy noted between the reported increase in CBV within the olfactory area (OLF) and the hemispheric analysis in previous Fig. S2e is due to the exclusion of one mouse from the FC dataset because of temperature-related artifacts. Unfortunately, we inadvertently failed to remove this mouse’s data from Fig. 1d, which led to the inconsistency. We sincerely apologize for this oversight. We have now corrected the dataset used in Fig. 1d to match the exclusion criteria applied in the FC analysis. The revised figure and corresponding text have been updated accordingly to ensure consistency across the manuscript.

Thank you again for helping us improve the clarity and accuracy of our work.

Reviewer #2 (Remarks to the Author):	
Comment	Response
(1) First, the authors' primary neurophysiological endpoint is very broad. Fifteen minutes of widespread changes in functional connectivity across the DMN (without careful statistical accounting for inter-animal variance) is not sufficient evidence that OFC-PV interneuron activation modulates DMN connectivity in a specific and reliable manner. Can the authors demonstrate that alternative interventions (e.g. modified interneuron activity in other brain regions) do not elicit similar broad changes? The existing control is not sufficient to isolate the level of specificity and selectivity necessary to highlight potential mechanism. This is especially critical given that the primary strength of the authors' use of fUS imaging is the high spatiotemporal resolution afforded by this approach in resolving microcircuit activity.	We appreciate the reviewer's concern regarding the specificity of our neurophysiological endpoint and the need for appropriate controls to isolate the effects of OFC-PV interneuron activation on DMN connectivity. To address this, we would like to highlight data from an independent, yet closely related, unpublished study conducted by a colleague within our research group. In this study, chemogenetic modulation of hippocampal interneurons was performed, followed by fUS imaging to assess brain-wide connectivity changes. Notably, this intervention did not result in widespread alterations across brain regions, nor did it produce any direct modulation of DMN connectivity. This data provides evidence that similar manipulations in other brain regions do not elicit the broad DMN-related effects observed with OFC-PV interneuron activation. We acknowledge the unpublished status of this data and respectfully request that it be treated with confidentiality. Fig: Franz et al. 1

CA2 pyramidal neurons in dorsal hippocampus of transgenic Amigo2-Cre mice were transduced with the inhibitory DREADD hM4Di-mCherry or mCherry only as control. The delivery took place via a viral Cre-dependent AAV-construct, bilaterally injected during early adulthood.

(A) functional connectivity of all measured brain regions in hM4Di+ mice relative to control after CNO application. A deficit exclusively limited to the DMN of mice is not detectable.

(2) Similarly, I am not convinced by the changes in the primary behavioral endpoints: social approach, social sniff, and object recognition and interaction tests. Is there any reason to believe that modifying interneuron activity in any region of the brain would not disrupt these general behaviors, even without DMN functional connectivity changes? Is there any reason to believe that these disruptions are selective to social behaviors or object memory (e.g. are the mice impaired across a range of non-social behavioral measures?).

Thank you for your thoughtful feedback. We agree that manipulating interneuron activity could potentially affect a broad range of behaviors. To address this concern, we included locomotion data as a representative non-social behavioral measure (line 121-126, We analysed locomotion data as a representative non-social behavioural measure in both NORT and social arena experiments. Our results show no significant differences in general locomotor activity between groups, suggesting that the observed changes in social behaviour are not due to global behavioural impairments nor motor deficits (Fig. S13). This implies that activating PV interneurons in the OFC causes both social and cognitive symptoms, as seen in patients with schizophrenia and AD). Our results show no significant differences in general locomotor activity between groups, suggesting that the observed changes in social behavior are not due to global behavioral impairments or motor deficits.

Furthermore, our study was designed to specifically investigate group-level social dynamics in a naturalistic setting, which differs from classical dyadic paradigms. This approach allows us to capture subtle and emergent social behaviors that may not be evident in standard tests.

	Additional support for the specificity of our findings comes from Stoller et al., (doi.org/10.1038/s41598-024-81930-w) that demonstrated region-specific modulation of social behavior without widespread behavioral disruption and from related work by Franz et al. (in submission), which also explores neuromodulatory effects on social behavior using similar paradigms, and. These studies reinforce the notion that targeted manipulation of interneuron activity can selectively influence social behavior without broadly impairing other behavioral domains¹⁰. We also revised the manuscript to interpret the cFos data more cautiously and avoid definitive claims regarding altered circuit output or excitation/inhibition balance.
--	--

Reviewer #3 (Remarks to the Author):	
Comment (1) The authors perform immunohistochemistry experiments to assess injection site and activation of hM3Dq receptors in OFC PV+ neurons. However, the images shown in the manuscript do not allow for proper assessment of the overall injection spread and comparison with the control group. In order to, evaluate the correct target of the OFC it would be helpful if authors include images at lower resolution that demonstrate correct targeting of the area. Furthermore, adding images for both groups	Response Confocal images with lower resolution demonstrating the injection sites for both groups are provided in Fig. S4-S5.

would help to visually compare expression patterns, in addition to the bar graph represented. Additional evaluation of expression in multiple mice and along also a frontal caudal axis via for example frequency mapping would strengthen the manuscript, but is not critical.	
(2) The authors assess the effectiveness of the chemogenetic activation via cFos staining and claim that the manipulation leads to increased firing of PV+ neurons which in turn causes higher inhibition of pyramidal cells and thereby generally lower excitatory output of the area. However, cFos staining alone does not provide adequate evidence of activation of the targeted neurons nor the effect on the local circuit, to support this claim. In order to, evaluate whether the manipulation produces the intended physiological effect and make definitive claims about increased inhibition and altered circuit output, in vivo electrophysiological experiments are necessary control experiments. Along the same line, the authors refer to E/I imbalance but do not show evidence of the presence of such an imbalance. This is rather problematic, in the absence of direct evidence of an imbalance, these statements are speculative and should be omitted.	We revised the discussion to more cautiously interpret the cFos data and have refrained from making definitive claims regarding altered circuit output or excitation/inhibition balance.
(3) The CNO dosage used in this study appears relatively high for this type of experiment (e.g., Gomez et al., 2017). It would be helpful if the authors add a justification for the chosen dose, as well as clarifications whether lower doses were tested and whether measurements for seizure like activity were conducted. While a	We acknowledge the reviewer's concern regarding the CNO dosage used in this study and added an explanation in the manuscript (Methods, lines 383-397, The CNO injection (5 mg/kg, 10 ml/kg injection volume) was conducted the following day, also at the beginning of the dark phase. The selected dose was based on pharmacokinetic and pharmacodynamic data presented in the study by Jendryka et al. (2019), which systematically compared the actions of clozapine-N-oxide, clozapine, and compound 21 in DREADD-based chemogenetics in mice ¹¹. Their findings indicate that CNO

pharmacokinetic analysis may not be necessary, any additional information on dose - response observations, or references to prior work using similar concentrations in comparable contexts, would help to support the validity of this approach.	exhibits a relatively low brain penetrance and requires higher systemic doses to achieve effective DREADD activation, particularly in acute experimental designs. In our study, CNO was administered as a single acute injection, and the timing of behavioural testing was aligned with its peak activity window. Lower doses were not tested in this experiment, as our chosen concentration falls within the range commonly used in similar behavioural studies and has been shown to reliably activate hM3Dq receptors without inducing off-target effects or seizure-like activity. No seizure-like behaviours were observed during or after administration, and animals were closely monitored throughout the testing period and one day after each set of experiments).
(4) In Figure 1 authors show CBV changes following their manipulation, however, in its current presentation the figure has a few issues that limit its clarity and interpretability. Below are some suggestions that may help the evaluation of this result. (4.1) In Figure 1c there are marked blobs outside of the brain mask, I suppose this might be a problem related to the registration process. I would recommend the authors verify correct registration and adjust the illustration.	Thanks for your accurate comment. A revised illustration is added to Fig 1.
(4.2) For the interpretation of Figure 1d an illustration with ROI locations as well as single data points, would be helpful.	To aid interpretation, an illustration of the ROI locations is provided in Fig. S6.
(4.3) The correlation presented in Figure 1f seems to include two outliers. It would be interesting to see the same correlation without these datapoints.	Thank you for pointing that out. To assess the influence of the two apparent outliers in Figure 1f, we recalculated the correlation after excluding them and updated the figure. The relationship between OFC and Insula remains strong, with a Pearson correlation coefficient of $r = 0.8400$, indicating that the association is robust and not solely dependent on those data points (Fig 1 is updated). The correlation graph including all data points is shown in Fig. S8 e-b.

(5) In Figure 2 FC alterations are displayed. The figure shows ingroup as well as between group comparisons which is very helpful for the interpretation of the result. However, starting from panel 2c onward, there appears to be a trend towards reduced connectivity also in the control group. The authors should elaborate on the potential effect of CNO in the absence of the DREADD receptor, especially considering the relatively high dosage used. It might also be worth to account for CNO-related effects in the analysis.

The reduced functional connectivity in control group is explained in **lines 88-96** (Another point is a trend toward reduced FC observed in the control group following CNO administration, raising the possibility that CNO itself—independent of DREADD expression—may contribute to altered connectivity. Supporting this interpretation, a recent study by Mantas et al. demonstrated that clozapine, the active metabolite of CNO, can affect brain-wide function, mediating changes in sensorimotor gating and connectivity patterns. In that study, functional ultrasound experiments demonstrated that clozapine administration led to a reduction in FC even in wild-type mice. These findings are consistent with our current observations and suggest that the FC decrease in the control group may reflect off-target effects of CNO-derived clozapine, potentially involving receptor interactions such as 5-HT_{2A}, D₂, and M1⁷)

An explanation regarding dosage used in these experiments is also mentioned in the revised version (Methods, **lines 383-397**, The CNO injection (5 mg/kg, 10 ml/kg injection volume) was conducted the following day, also at the beginning of the dark phase. The selected dose was based on pharmacokinetic and pharmacodynamic data presented in the study by Jendryka et al. (2019), which systematically compared the actions of clozapine-N-oxide, clozapine, and compound 21 in DREADD-based chemogenetics in mice ¹¹. Their findings indicate that

	CNO exhibits a relatively low brain penetrance and requires higher systemic doses to achieve effective DREADD activation, particularly in acute experimental designs. In our study, CNO was administered as a single acute injection, and the timing of behavioural testing was aligned with its peak activity window. Lower doses were not tested in this experiment, as our chosen concentration falls within the range commonly used in similar behavioural studies and has been shown to reliably activate hM3Dq receptors without inducing off-target effects or seizure-like activity. No seizure-like behaviours were observed during or after administration, and animals were closely monitored throughout the testing period and one day after each set of experiments).
(6) In Figure S3 authors show FC matrix for each group, however in its current form the representation and statistical approach used are not entirely clear. If each matrix shows baseline vs. post-CNO comparison within one group, it appears that CNO administration leads to FC alterations in the control group. The authors should elaborate on this finding, especially considering the rather high CNO dosage and its potential off-target effects. In addition, such a within group comparison should be analyzed using paired statistical testing, since it refers to repeated measures within the same animals. Lastly, adding a matrix that demonstrates the between group differences in baseline as well as post-CNO window would enhance the interpretation of these effects. Alternatively, if the matrix shows alterations within group but the significance markers reflect between group comparisons, unpaired statistics are appropriate, but within group statistical tests would be missing. The addition of these comparisons would be beneficial to demonstrate that CNO does not affect FC in control animals and that	Thank you for this important observation. We agree that the trend toward reduced functional connectivity (FC) observed in the control group, particularly from Figure 2c onward, raises the valid concern that CNO itself—independent of DREADD expression—may contribute to altered connectivity. Indeed, supporting this interpretation, a recent study by Mantas et al. (2025) demonstrated that clozapine, the active metabolite of CNO, affects brain-wide function, mediating changes in sensorimotor gating and connectivity patterns (Neuropsychopharmacology, https://www.nature.com/articles/s41386-025-02060-z). In that study, fUS experiments were conducted at our site by a former PhD student, showing that clozapine administration led to a reduction in functional connectivity, even in wild-type mice not expressing DREADDs. These results are in line with our current observations of FC reduction in the control group following CNO injection. Taken together, these data suggest that the observed FC decrease in the control group may reflect off-target effects of CNO-derived clozapine, consistent with previously reported receptor interactions (e.g., 5-HT2A, D2, M1). We now acknowledge this possibility in the revised manuscript and have added a paragraph discussing the potential contribution of CNO-related effects (see lines 88-96, Another point is a trend toward reduced FC observed in the control group following CNO administration, raising the possibility that CNO itself—independent of DREADD expression—may contribute to altered connectivity. Supporting this interpretation, a recent study by Mantas et al. demonstrated that clozapine, the active metabolite of CNO, can affect brain-wide function, mediating changes in sensorimotor gating and connectivity patterns. In that study, functional ultrasound experiments demonstrated that clozapine administration led to a reduction in FC even in wild-type mice. These findings are consistent with our current observations and

the baseline connectivity is comparable between groups.	suggest that the FC decrease in the control group may reflect off-target effects of CNO-derived clozapine, potentially involving receptor interactions such as 5-HT_{2A}, D₂, and M1⁷).
(7) Authors mention how activation of OFC PV+ neurons leads to CBV and connectivity changes in various areas of the brain. Further elaboration on the significance and rationale of these areas and the results itself would add value and help contextualize the findings. Below some examples where this could be beneficial. (7.1) Line 46 onwards. Authors mention areas that show CBV increase, providing a rationale for why these specific areas show the effect would aid the interpretation of the result.	We thank the reviewer for this insightful comment. In response, we have expanded the relevant section of the manuscript to provide a mechanistic rationale for the observed CBV increases in the prelimbic cortex (PreL), primary motor cortex (M1), and secondary motor cortex (M2) following chemogenetic activation of PV interneurons in the OFC. Specifically, PV interneurons are known to regulate cortical network synchrony through gamma oscillations, which are critical for coordinating activity across distant brain regions. Their activation enhances the precision of local excitatory output via feedforward and feedback inhibition, thereby facilitating more effective communication with anatomically connected areas such as PrL, M1, and M2. These regions are part of known OFC-centered circuits involved in top-down modulation of cognitive and motor functions. The increased CBV in these areas likely reflects enhanced metabolic demand due to this coordinated network activity¹²⁻¹⁵. We hope this addition clarifies the rationale behind our findings and strengthens the interpretation of the data (see lines 136-148, Our preclinical study supports this idea by showing that decreased FC in DMN regions, particularly in the hippocampus, is connected to social impairment. The observed CBV increases in the PreL, M1, and M2 following chemogenetic activation of PV interneurons in the OFC likely reflect enhanced functional connectivity and network-level modulation. PV interneurons are known to orchestrate cortical network synchrony, particularly through gamma oscillations, which facilitate long-range communication between brain regions^{13,16}. Their activation sharpens local excitatory output via feedforward and feedback inhibition, increasing the signal-to-noise ratio and potentially enhancing downstream activation in anatomically connected areas¹⁴. Given the established anatomical and functional connectivity between the OFC and medial prefrontal as well as motor cortices¹⁵, the increased CBV in PrL, M1, and M2 may reflect a coordinated network response to PV-mediated modulation of OFC output.).
(7.2) Line 53 onwards. Similarly, for the reported reductions in FC, further elaboration on the specific brain regions and the potential relevance of these alterations would help readers understand the implications of these findings.	We have addressed this point in the revised manuscript (lines 156-174, The observed reductions in FC between DMN regions following chemogenetic activation of OFC PV interneurons likely reflect changes in local circuit dynamics that propagates through large-scale brain networks. PV interneurons are fast-spiking GABAergic cells that exert strong perisomatic inhibition on pyramidal neurons, thereby regulating cortical output. While

	chemogenetic activation of these interneurons in the OFC is expected to influence local circuit dynamics, the precise impact on overall OFC activity remains to be fully determined. Nevertheless, such modulation may alter the excitatory drive to downstream and functionally connected regions such as the hippocampus, thalamus, retrosplenial cortex, and anterior cingulate cortex. This mechanism is consistent with prior findings showing that PV interneuron activity can modulate long-range network synchrony and memory-related processes by gating pyramidal neuron output¹⁷. Moreover, the OFC plays a critical role in top-down modulation of limbic and associative cortices. Disruption of this regulatory influence through enhanced local inhibition may desynchronize activity across the DMN, leading to reduced temporal coherence and FC. Similar network-level effects have been observed in studies where PV interneuron activation in prefrontal regions altered fear memory expression and disrupted connectivity with subcortical targets^{18,19}. These findings suggest that PV interneuron-mediated inhibition not only shapes local circuit dynamics but also exerts widespread influence on distributed cognitive networks).
(7.3) Line 61 onwards. Authors should elaborate on the rationale behind the hemisphere-specific analysis of FC.	We have addressed this point in the revised manuscript (lines 75–86, Hemisphere-specific analysis was performed in our study to account for known hemispheric asymmetries and functional lateralization in brain organization. Functional lateralization refers to the preferential engagement of one hemisphere in specific cognitive or neural processes, a phenomenon well-documented in both humans and rodents. Additionally, hemispheric asymmetries in mice—particularly in cortical and subcortical connectivity—have been increasingly recognized, supporting the relevance of lateralized analyses in preclinical models^{20–22}. Beyond biological factors, technical considerations may also contribute to apparent lateralization. For example, if signal quality is compromised on one side due to suboptimal gel coupling or increased air content in the skull, this can lead to systematic differences in connectivity at the group level. Including hemisphere-specific analyses helps ensure that both biological and technical sources of asymmetry are appropriately accounted for).
(7.4) Line 65 onwards. Authors report that against the general trend of decreased FC, some areas also show increased FC. Further elaboration on the relevance of this result would add value to the finding.	We have addressed this point in the revised manuscript (lines 181–197, Although the dominant network-level effect of OFC PV-interneuron activation was a reduction in functional connectivity across canonical DMN nodes, the pattern was not uniformly suppressive — a subset of region pairs showed increased FC. These increases may result from compensatory recruitment of alternative circuits when DMN-mediated integration is weakened or disinhibition of downstream or parallel pathways caused by altered local balance in the OFC. Such reconfiguration is consistent with the OFC’s role as a top-down regulator. These

	increases are best viewed not as random noise but as a systematic reconfiguration: when OFC PV-interneuron activation desynchronizes DMN hubs, the brain can both (A) recruit alternative sensory- and limbic-driven pathways (compensation) and (B) disinhibit circuits that are normally held in check by OFC output, producing localized hyperconnectivity. Whether such increases are adaptive (restoring function) or maladaptive (producing aberrant synchrony) will depend on behavioral outcome measures and whether the same increases are sustained or transient. Functionally, one would predict that increases involving limbic-sensory axes (e.g., amygdala→sensory cortex, piriform→perirhinal) would bias processing toward externally salient or emotionally laden stimuli, whereas increases involving motor and parietal nodes (e.g., M2↔RSP/SS) would favor sensorimotor re-engagement).
(8) For fUs experiments authors state that heart rate, and other physiological parameters were monitored, including a supplementary illustration of these measures would enhance interpretation of the results.	Fig. S7 shows heart rate and temperature of animals during fUS experiment.
(9) In order to, measure social behavior authors placed animals in an automated tracking apparatus and measured naturalistic behavior over multiple days. Further explanation of this approach along with references to previous studies using similar methods would help contextualize these results. Specifically, clarifications regarding the following methodological details would facilitate interpretation of the results.	The concept of studying animals in more naturalistic experimental settings has gained traction over the past decade. For instance, Shemesh et al. (2013) introduced the “Social Box,” an automated phenotyping system using video-based color recognition to track multiple mice in a semi-natural environment designed by the Alon Chen group at the Max Planck Institute of Psychiatry²³. This setup enabled the quantification of social behaviors without manual scoring and demonstrated how individual behavior is shaped by group dynamics. Building on this, Forkosh et al. (2019) used the same system to identify stable personality traits in mice²⁴. These and other studies have successfully employed behavioral setups involving four or more mice simultaneously, highlighting the feasibility and scientific value of multi-animal tracking in controlled environments. Approaches such as our system rely on physical markers (in our setup RFID tags). Recent advances in machine learning and computer vision have enabled tracking of multiple animals and body parts. In such setups, mice develop interdependent spatial and behavioral patterns that cannot be explained by individual preferences alone—a phenomenon referred to as behavioral convergence. These emergent group-level dynamics are sensitive to social composition and environmental context. Therefore, combining hM3Dq and control mice in the same group for long-term behavioral assays could obscure the effects of our manipulation by introducing artificial social hierarchies or

	masking group-level adaptations. Our design ensures that the observed behavioral changes reflect the neuronal manipulation rather than unintended social confounds. We have addressed this point in the revised manuscript (lines 198-209, Our behavioural setup, as well as other group based behavioural setups such as shemesh et al, involves four mice interacting freely within a semi-naturalistic arena. In such environments, mice exhibit interdependent spatial and behavioural patterns that cannot be attributed to individual preferences alone—a phenomenon known as behavioural convergence. These emergent group-level dynamics are highly sensitive to both social composition and environmental context^{10,23-29}. Social behaviour in group-living species like mice is inherently complex and context-dependent. Traditional paradigms, while informative, may not capture the higher-order interactions and emergent properties that arise in group settings. Our approach—housing either all hM3Dq or all control mice together—was specifically chosen to examine how manipulation influences collective behaviour while minimizing potential confounds such as behaviour adaptation which has been observed in mixed groups of animals^{30,31}).
(9.1) Social behavior was assessed within groups of either all hM3Dq or all control mice. However, in many established social behavior paradigms, interactions are typically measured between manipulated and non-manipulated control animals. A discussion regarding the chosen experimental design and possible confounding factors should be added. Moreover, including a second measure of social behavior with a manipulated vs. non manipulated control animal comparison would strengthen the conclusions and help rule out group-level behavioral confounds.	We thank the reviewer for this important point, which is added to manuscript (lines 198-209, Our behavioural setup, as well as other group based behavioural setups such as shemesh et al, involves four mice interacting freely within a semi-naturalistic arena. In such environments, mice exhibit interdependent spatial and behavioural patterns that cannot be attributed to individual preferences alone—a phenomenon known as behavioural convergence. These emergent group-level dynamics are highly sensitive to both social composition and environmental context^{10,23-29}. Social behaviour in group-living species like mice is inherently complex and context-dependent. Traditional paradigms, while informative, may not capture the higher-order interactions and emergent properties that arise in group settings. Our approach—housing either all hM3Dq or all control mice together—was specifically chosen to examine how manipulation influences collective behaviour while minimizing potential confounds such as behaviour adaptation which has been observed in mixed groups of animals^{30,31}). Indeed, many classical paradigms in social behavior research rely on dyadic interactions between manipulated and non-manipulated animals to isolate specific social responses. However, our study was designed to address a complementary question: how neuronal manipulation affects emergent group-level social dynamics in a more naturalistic setting, as well as individual behaviors. Social behavior in group-living species like mice is inherently complex and context-dependent. Traditional paradigms, while

	informative, may not capture the higher-order interactions and emergent properties that arise in group settings. Our approach—housing either all hM3Dq or all control mice together—was specifically chosen to examine how manipulation influences collective behavior while minimizing potential confounds such as behavior adaptation which has been observed in mixed groups of manipulated and non-manipulated animals.
(9.2) Adding a justification for the duration of the habituation period with references to previous studies would enhance clarity.	The duration of the habituation period in our study was informed by previous research demonstrating that both motor activity and social behaviors tend to decline during the initial days following introduction to a new environment. Studies such as Peleh et al. (2019) and Kas et al. (2009) have shown that mice exhibit elevated exploratory and social behaviors immediately after placement in a novel arena, which gradually stabilize over approximately three days. This habituation period is therefore critical to minimize novelty-induced behavioral artifacts and ensure that subsequent measurements reflect baseline, context-adapted behavior ^{26,32} (lines 372-379, After a three-day habituation phase in the arena (including one day baseline), during which mice were monitored. The duration of the habituation period in our study was informed by previous research demonstrating that both motor activity and social behaviours tend to decline during the initial days following introduction to a new environment. Studies such as Peleh et al. (2019) and Kas et al. (2009) ^{26,27,32} have shown that mice exhibit elevated exploratory and social behaviours immediately after placement in a novel arena, which gradually stabilize over approximately two to three days).
(9.3) The method used to score social behavior, including the definition of what constitutes social approach or social sniffing, should be described in greater detail.	We have addressed this point in the method part of revised manuscript (see lines 505-512, Social sniffing was defined as an interaction occurring when the distance between two mice was less than 3.5 cm and sustained for a minimum duration of 0.33 seconds. Social approach was characterized by a combination of spatial and movement criteria: the approaching mouse (mouse 1) had to move toward the other mouse (mouse 2) from a distance of less than 100 cm, with a movement angle of less than 45°, covering a minimum distance of 7 cm at a velocity exceeding 4 cm/s. These criteria were selected to ensure consistent and objective identification of social interactions across trials.²⁶).
(9.4) Further elaboration on the rationale behind the vehicle vs. CNO approach (rather than baseline	Thank you for raising this important point. We chose to include the vehicle condition as a double control to ensure that any observed behavioral changes following CNO administration were not simply due to the stress of injection, handling,

vs. post-CNO) would enhance understanding.	or removal from the social arena. This design allows us to isolate the specific effects of chemogenetic activation from general procedural influences. While our primary comparisons focus on Vehicle vs. CNO, we also collected baseline data during the third day of the habituation phase, when animals were already familiar with the environment but had not yet received any injections. As an example we include one supplementary figure (Fig S12) showing consistent differences in social approach behavior when comparing baseline vs. CNO and Vehicle vs. CNO during the 12-hour dark phase, supporting the specificity of the CNO effect beyond general handling or injection stress.
(9.5) Further explanation on data representation should be added in order to facilitate comprehension of the results. For example, are single data points individual bouts of behavior or per-animal averages? Clarifying this would help evaluate the variability and consistency of the effect across animals. If possible, displaying individual mouse data could further strengthen the analysis.	We have addressed this point in the method part of revised manuscript (lines 513-515, Each data point represents the average behaviour per hour per mouse. Mean values were calculated across the entire monitoring phase).
(10) In order to investigate whether alterations in social behavior were specific to this behavioral domain or rather reflect a reduction in exploratory behavior, the authors performed a NOR task. Further explanations on this task and the resulting effect would give additional value to this finding.	We have addressed this point in supplementary Fig. S13 showing that locomotion activity in social arena experiment and NORT is not affected by our modulation.
(10.1) Further elaboration on the choice of the 1-hour interval between training and testing session would be helpful. Offering a rationale for this duration, as opposed to longer intervals commonly used, would provide valuable context.	We chose a protocol with minimum habituation sessions as explained by Leger et al. 1-hour interval is used to assess short-term memory, providing sufficient time for memory consolidation^{33,34}. Additionally, this timing aligns with the pharmacokinetic profile of clozapine-N-oxide (CNO), which we administered acutely as a single injection prior to the training session. We have addressed this point in the method part of revised manuscript (see lines 407-417, The novel object recognition test (NORT) was conducted over two days, with a one-hour delay post-training, based on a modified version of the protocol from Lueptow³⁵. The protocol was designed with limited habituation sessions, in line with recommendations by Leger et al. A 1-hour interval between training and testing was chosen to assess short-term memory, allowing sufficient time for memory

	consolidation^{33,34}. This timing also corresponds with the pharmacokinetic profile of clozapine-N-oxide (CNO), which was administered acutely prior to the training session. Additionally, this design helps reduce potential confounds related to drug metabolism and behavioural variability. Prior to the NORT, the social arena was modified by removing nests, food hoppers, and water bottles to minimize external distractions (Fig. S14). 20 mice from cohort one were used for the NORT experiment).
(10.2) The method used to measure and score object exploration should be described in greater detail.	Thank you for your comment. We have clarified the method used to measure and score object exploration in the NORT in lines 418-425 (During both sessions, object exploration was manually scored based on video recordings. Exploration was defined as the mouse directing its nose toward the object at a distance of ≤ 2 cm and/or actively sniffing or touching the object. Merely passing by the object without investigative behaviour was not counted as exploration. Each session lasted 5 minutes, but if a mouse did not reach at least 20 seconds of combined exploration time for both objects within that period, the session was extended beyond 5 minutes until the 20-second criterion was met. If the mouse failed to reach the 20-second minimum within 10 minutes, the animal was excluded from further analysis, as insufficient exploration time would not allow reliable assessment of learning or discrimination)
(10.3) The authors mention that 4 mice did not meet the cut off duration due to a lack of active periods. Considering the group sizes (n=9 and n=7), the exclusion of 4 mice seems substantial. Adding a discussion regarding the potential impact of this exclusion on the analysis would enhance clarity.	We appreciate the reviewer's observation regarding the exclusion of 4 mice due to insufficient object exploration time. As noted, mice were required to reach a minimum of 20 seconds of total exploration across both objects during both sessions. This threshold was set to ensure that animals had sufficient interaction with the objects to support reliable assessment of recognition memory. The exclusion of 4 mice reduced the final group sizes to n = 9 and n = 7. While this reduction is notable, we believe it was necessary to maintain the validity and interpretability of the behavioral data. Including animals that did not meet the exploration threshold could introduce variability unrelated to memory performance and potentially obscure true effects (lines 425-432, The final results involved 9 hM3Dq mice versus 7 control mice. 4 mice were excluded from the analysis as they did not meet the cut-off duration criteria due to a lack of active periods during the experiment. This exclusion was necessary to ensure data quality and interpretability. Including animals with minimal engagement could have introduced variability unrelated to memory performance. Nonetheless, we are aware that the reduced group size may limit statistical power and should be considered when interpreting the results)
(10.4) Although not necessary, additional controls assessing	Figure S13 provides locomotion activity in both NORT and social arena experiment.

general locomotor activity (even as measured within the same task) would help rule out confounding effects. Similarly, if the authors would like to discuss cognitive impairments more broadly, including further assessments of other types of memory or other cognitive domains could strengthen and expand the conclusions, though this is not essential and might be outside the scope of this work.	
(10.5) It would be helpful if the authors could expand the discussion of the observed memory impairments, including possible underlying mechanisms and their relevance to the study's overall conclusions. Additionally, considering how these memory impairments might relate to or influence the reductions in social behavior could provide further valuable insight.	We have addressed this point in the revised manuscript (lines 217–244, We also observed memory impairment, which may be linked to altered FC within the hippocampus and its associated networks. Specifically, the observed reductions in FC between DMN regions likely reflect disrupted integration of cognitive and affective processes. Enhanced inhibition in the OFC may have mechanistically desynchronized pyramidal output, thereby disrupting long-range coordination across these mnemonic circuits¹⁶. The dorsal hippocampus and subiculum are central to episodic memory and spatial navigation, and their reduced connectivity with the thalamus and somatosensory areas may impair hippocampal-cortical communication essential for contextual processing. Chemogenetic activation of OFC PV interneurons disrupted connectivity within hippocampal–thalamic–retrosplenial–cingulate loops that are central to episodic and contextual memory. Reduced FC between the dorsal hippocampus and subiculum, thalamus, and somatosensory areas likely impaired hippocampal–cortical communication essential for contextual integration^{36,37}. The thalamus, acting as a relay hub and a critical relay for aligning hippocampal activity with cortical representations, supports attentional modulation and consciousness, and its weakened connections with the RSP and ACg (two DMN hubs) suggest impaired sensory-cognitive integration^{38,39}. The human retrosplenial cortex, involved in autobiographical memory and scene construction, and the anterior cingulate cortex, implicated in emotional regulation and cognitive control, both showed reduced connectivity in our study that may underlie deficits in internally directed thought and top-down regulation^{40,41}. Disruptions in DMN connectivity, particularly involving the hippocampus, have been associated with cognitive deficits, including memory impairment⁴². Thus, our findings link local inhibitory modulation in the OFC to large-scale network alterations that undermine hippocampal-dependent memory, mirroring clinical observations in disorders such as schizophrenia and Alzheimer's disease where hippocampal–

	DMN dysconnectivity contributes to cognitive decline⁴³. Our findings also mirror clinical observations of a relationship between social impairment and cognitive decline and contribute to elucidating the mechanism underlying this association⁴⁴⁻⁴⁶).
(11) The authors describe how altering OFC PV activity affects both connectivity and behavior. In order to, link these two main results, the authors might consider exploring this relationship between connectivity and behavioral alterations using for example multivariate modelling approaches. While this could provide valuable insights, the use of different cohorts for the two measures might make this a difficult endeavor and such an analysis is not essential.	Thank you for this insightful suggestion. This idea was discussed with consulting statisticians. However, as you already mentioned, we were advised that such a multivariate analysis would only be possible from a methodological point of view if both endpoints would have been measured on the same subjects, or at least on a subset of subjects from both cohorts. Otherwise, it is not possible, to estimate the correlation between the two endpoints and we were advised that two separate analyses are the most suitable way to proceed (Line 263-269, Third, FC and behaviour were measured in separate cohorts, preventing within-subject correlations that could directly link connectivity changes to individual outcomes. Our behavioural assays also focused primarily on sociability and recognition memory, leaving other domains such as anxiety-like behaviour, reward sensitivity, and cognitive flexibility unexplored. Fourth, although DREADDs enable targeted manipulation, systemic CNO can convert to clozapine and influence endogenous receptors. Use of next-generation ligands could reduce this confound).

1. Grandjean, J. et al. *Nat. Neurosci.* **26**, 673–681 (2023).
2. Grandjean, J., Schroeter, A., Batata, I. & Rudin, M. *NeuroImage* **102**, 838–847 (2014).
3. Bukhari, Q., Schroeter, A., Cole, D.M. & Rudin, M. *Front. Neural Circuits* **11**, 5 (2017).
4. Mandino, F., Vujic, S., Grandjean, J. & Lake, E.M.R. *Cereb. Cortex* **34**, bhad478 (2023).
5. Lu, H. et al. *Proc. Natl. Acad. Sci.* **109**, 3979–3984 (2012).
6. Sforazzini, F., Schwarz, A.J., Galbusera, A., Bifone, A. & Gozzi, A. *NeuroImage* **87**, 403–415 (2014).
7. Mantas, I. et al. *Neuropsychopharmacology* **50**, 721–730 (2025).
8. Vo, T.T. et al. *Proc. Natl. Acad. Sci.* **120**, e2220777120 (2023).
9. Anenberg, E., Chan, A.W., Xie, Y., LeDue, J.M. & Murphy, T.H. *J Cereb Blood Flow Metabolism* **35**, 1579–1586 (2015).
10. Stoller, F. et al. *Sci. Rep.* **14**, 30492 (2024).
11. Jendryka, M. et al. *Sci Rep-uk* **9**, 4522 (2019).
12. Sohal, V.S., Zhang, F., Yizhar, O. & Deisseroth, K. *Nature* **459**, 698–702 (2009).
13. Cardin, J.A. et al. *Nature* **459**, 663–667 (2009).
14. Tremblay, R., Lee, S. & Rudy, B. *Neuron* **91**, 260–292 (2016).
15. Barbas, H. *J. Anat.* **211**, 237–249 (2007).
16. Sohal, V.S. & Rubenstein, J.L.R. *Mol Psychiatr* **24**, 1248–1257 (2019).
17. Hu, N.-Y. et al. *Transl. Psychiatry* **15**, 243 (2025).
18. Marín, O. *Eur. Neuropsychopharmacol.* **82**, 44–52 (2024).
19. Yang, S.-S., Mack, N.R., Shu, Y. & Gao, W.-J. *Front. Neural Circuits* **15**, 716408 (2021).
20. Karenina, K., Giljov, A., Ingram, J., Rowntree, V.J. & Malashichev, Y. *Nat. Ecol. Evol.* **1**, 0030 (2017).
21. Rivera-Olvera, A. et al. *Mol. Psychiatry* **30**, 489–496 (2025).
22. Ocklenburg, S., Basbasse, Y.E., Ströckens, F. & Müller-Alcazar, A. *Commun. Biol.* **6**, 521 (2023).

23. Shemesh, Y. et al. *eLife* **2**, e00759 (2013).
24. Forkosh, O. et al. *Nat. Neurosci.* **22**, 2023–2028 (2019).
25. Weissbrod, A. et al. *Nat. Commun.* **4**, 2018 (2013).
26. Peleh, T., Bai, X., Kas, M.J.H. & Hengerer, B. *J Neurosci Meth* **325**, 108323 (2019).
27. Peleh, T. et al. *Neuroscience* **445**, 95–108 (2020).
28. Ike, K.G.O. et al. *Mol. Psychiatry* **29**, 518–528 (2024).
29. Ike, K.G.O., Boer, S.F. de, Buwalda, B. & Kas, M.J.H. *Neurosci. Biobehav. Rev.* **116**, 251–267 (2020).
30. Hill, K. & Boyd, R. *Science* **371**, 235–236 (2021).
31. Claidière, N. & Whiten, A. *Psychol. Bull.* **138**, 126–145 (2012).
32. Kas, M.J.H. et al. *Genes, Brain Behav.* **8**, 13–22 (2009).
33. Jurdak, N. & Kanarek, R.B. *Physiol. Behav.* **96**, 1–5 (2009).
34. Leger, M. et al. *Nat. Protoc.* **8**, 2531–2537 (2013).
35. Lueptow, L.M. *J. Vis. Exp.* (2017).doi:10.3791/55718
36. Aggleton, J.P. et al. *Eur. J. Neurosci.* **31**, 2292–2307 (2010).
37. Eichenbaum, H. *Nat. Rev. Neurosci.* **18**, 547–558 (2017).
38. Vann, S.D., Aggleton, J.P. & Maguire, E.A. *Nat. Rev. Neurosci.* **10**, 792–802 (2009).
39. Leech, R. & Sharp, D.J. *Brain* **137**, 12–32 (2014).
40. Fallon, N., Chiu, Y., Nurmikko, T. & Stancak, A. *PLoS ONE* **11**, e0159198 (2016).
41. Sharaev, M.G., Zavyalova, V.V., Ushakov, V.L., Kartashov, S.I. & Velichkovsky, B.M. *Front. Hum. Neurosci.* **10**, 14 (2016).
42. Wang, J., Liu, S., Liang, P., Cui, B. & Wang, Z. *Brain Commun.* **7**, fcae476 (2024).
43. Brier, M.R. et al. *J. Neurosci.* **32**, 8890–8899 (2012).
44. Kuiper, J.S. et al. *Int. J. Epidemiology* **45**, 1169–1206 (2016).
45. Kelly, M.E. et al. *Syst. Rev.* **6**, 259 (2017).

46. Piolatto, M. et al. *BMC Public Heal.* **22**, 278 (2022).

Reviewer #3 (Remarks to the Author)	
Comment	Response
1. The addition of lower-resolution confocal images illustrating viral expression is appreciated. However, the overall image quality could still be improved. In Figure S5 (hM3Dq group), the extent and intensity of viral expression appear somewhat variable across animals. It might be helpful to briefly comment on this variability or provide a summary figure showing expression extent across animals.	We thank the reviewer for this comment. Viral expression exhibited some inter-animal variability, which is in the expected range and likely reflects minor differences in injection spread within the orbitofrontal cortex (OFC). In mice, the OFC extends from approximately +3.2 mm to +1.98 mm relative to bregma, and our target coordinate (+2.6 mm) lies within this range. Despite modest rostro-caudal variability, viral expression consistently covered the intended OFC subregions.
2. While the authors did not provide the electrophysiological control experiments previously suggested, I appreciate their revised and more cautious interpretation of the c-Fos data, as well as their restraint from making definitive claims regarding altered circuit output or excitation/inhibition balance. 2.1 In this context, the authors should note that the legend of Figure S14 still mentions an “E/I imbalance,” which should be revised for consistency.	Thank you for pointing this out. The legend of Figure S14 has been revised for consistency, and the reference to ‘E/I imbalance’ has been removed.
2.2 Additionally, Figure 1a schematically depicts	Thank you for the suggestion. Figure 1a and its legend have been revised to remove detailed circuit-level

the effect of PV⁺ neuron activation on local pyramidal neurons. Since the manuscript does not provide experimental data demonstrating how this manipulation affects other neuronal populations, I recommend refraining from detailed circuit-level interpretations in the schematic and corresponding legend.	interpretations, ensuring consistency with the experimental scope.
3. I appreciate the authors' detailed explanation regarding the chosen CNO dosage, and especially the note that animals were carefully monitored and did not exhibit seizure-like behavior. However, the justification based on Jendryka et al. (2019) is not entirely adequate. The study by Jendryka et al. does not report specific data for a 5 mg/kg dose, and although the authors suggest that dose ranges from 3 to 5 mg/kg CNO should be effective, the study explicitly refers to inhibitory DREADD receptors (hM4Di), not excitatory hM3Dq receptors as used here. Jendryka et al. further emphasize that receptor type significantly influences effective	Thank you for this important comment. We have added two additional references supporting the use of 5 mg/kg CNO for excitatory hM3Dq activation, as noted in lines 388–389 of the revised manuscript

dosing. I therefore suggest that the authors reference additional studies supporting the chosen dose for excitatory DREADDs, or provide further clarification.	
4. I thank the authors for their added explanation regarding the FC alterations observed in the control group (Figure S9). 4.1 However, the authors did not address my previous comment regarding the use of paired statistical tests for within-group comparisons. 4.2 In addition, including a connectivity matrix that depicts between-group differences both at baseline and post-CNO would substantially enhance interpretability.	We thank the reviewer for identifying this inconsistency in the text, which escaped our attention. All statistical analyses were in fact performed using paired tests, not unpaired tests as incorrectly stated. 4.2, we fully appreciate the reviewer's interest in a connectivity matrix showing between-group differences at baseline and post-CNO. However, because baseline connectivity patterns already differ across animals in both hM3Dq and control groups, making between-group comparisons of absolute connectivity uninformative in this dataset. To address this transparently, we now provide a single heatmap showing baseline connectivity patterns of both groups side-by-side. This presentation highlights the substantial inter-individual variability present already at baseline, and clarifies why direct matrix subtraction or statistical between-group comparison are not appropriate.

Moreover, robust between-group connectivity analyses would require a substantially larger sample size to account for individual variability. Increasing the sample size was not feasible due to adherence to the 3R principles (Reduction and Refinement) and institutional animal-use constraints.

4.3 Lastly, in Figure S9 the “circled stars” denoting FDR-corrected results are barely visible; this might be a resolution issue that could be easily improved.

High resolution figures will be provided for publication.

5. The expanded explanation of the social behavioral measurements is very helpful and greatly improves clarity.
5.1 The rationale for the chosen test is now clearly stated. Nevertheless, given that social behavior is

We appreciate this thoughtful suggestion. While incorporating a classical social interaction test was beyond the scope of the present study, we agree that such an approach could provide complementary insights into reciprocal aspects of social behavior and represents an excellent avenue for future research.

inherently reciprocal, it could be informative to complement the current paradigm with a classical social interaction test involving an unaffected stimulus mouse. Although such tests have limitations, their inclusion could provide additional insight into different aspects of social behavior.	
5.2 Similarly, while adding further memory-related tests may not be feasible, doing so would strengthen the claims regarding memory impairments.	While incorporating additional memory-related tests was not feasible within the scope of the current study, we agree that doing so would strengthen the conclusions regarding memory impairments and represents an excellent direction for future research.
5.3 Baseline data for social approach were appropriately added in Figure S12; however, equivalent baseline data for social sniffing are still missing and should be included.	Figure S12 is updated.
5.4 While vehicle versus CNO comparisons are relevant, including analyses comparing the DREADD and control groups under the various conditions would add valuable context. For instance, a mixed linear model could be employed to appropriately test these interactions.	We thank the reviewer for the suggestion. As recommended, we performed a two way between subjects ANOVA (Dunnett's T3 multiple comparisons test) (supplementary file Table S1-S2).
5.5 I appreciate the inclusion of locomotion measures as behavioral controls. However, Figure S13 should	We adjusted the figure as well as the legend.

display individual data points, and the legend should be corrected to avoid indicating statistical significance where none is present.	
5.6 Finally, the explanation for the exclusion of four mice from the NOR test is appreciated, but a brief discussion addressing why several animals did not reach the inclusion threshold would provide useful context, though this is only a minor suggestion.	We thank the reviewer for this suggestion. The reason for excluding the four mice from the NOR test is explained in detail in lines 422–425 of the manuscript.
It may be beneficial to further moderate statements implying mechanistic or causal relationships, particularly regarding potential treatment implications. Given the correlational design and use of separate experimental cohorts, more cautious language would align well with the data and maintain a balanced interpretation.	Thank you for the comment. We have further toned down all causal language to ensure our interpretation matches the correlational design (lines 222-251).